# HPOD: **Hyperparameter Optimization for Unsupervised Outlier Detection**

**Yue Zhao**[1]   **Leman Akoglu**[2]

[1]University of Southern California
[2]Carnegie Mellon University

**Abstract**   Given an unsupervised outlier detection (OD) algorithm, how can we optimize its hyperparameter(s) (HP) on a *new dataset, without using any labels*? In this work, we address this challenging hyperparameter optimization for unsupervised OD problem, and propose *the first continuous HP search method* called HPOD. It capitalizes on the prior performance of a large collection of HPs on *existing* OD benchmark datasets, and transfers this information to enable HP evaluation on a new dataset without labels. Also, HPOD adapts a prominent, (originally) supervised, sampling paradigm to efficiently identify promising HPs in iterations. Extensive experiments show that HPOD works for both deep (e.g., Robust AutoEncoder (RAE)) and shallow (e.g., Local Outlier Factor (LOF) and Isolation Forest (iForest)) algorithms on discrete and continuous HP spaces. HPOD outperforms a wide range of diverse baselines with 37% improvement on average over the minimal loss HPs of RAE, and 58% and 66% improvement on average over the default HPs of LOF and iForest.

## 1 Introduction

Although a long list of unsupervised outlier detection (OD) algorithms have been proposed (Aggarwal, 2013; Campos et al., 2016; Pang et al., 2021), how to optimize their hyperparameter(s) (HP) remains underexplored. Without hyperparameter optimization (HPO) methods, practitioners often use the default HP of an OD algorithm, which is hardly optimal given many OD algorithms are sensitive to HPs. For example, a recent study by Zhao et al. (2021) reports that by varying the number of nearest neighbors in local outlier factor (LOF) (Breunig et al., 2000) while fixing other conditions, up to 10× performance difference is observed in some datasets. The literature also shows that HP sensitivity is exacerbated for deep OD models with more 'knobs' (e.g., HPs and architectures) (Ding et al., 2022), which we also observe in this study—deep robust autoencoder (RAE) (Zhou and Paffenroth, 2017) exhibits up to 37× performance variation under different HPs.

In supervised learning, one can use ground truth labels to evaluate the performance of an HP, including grid and random search (Bergstra and Bengio, 2012) as well as more efficient Sequential Model-based Bayesian Optimization (SMBO) (Jones et al., 1998). Unlike the simple methods, SMBO builds a cheap regression model (called the surrogate function) of the expensive objective function (which often requires ground truth labels), and uses it to iteratively select the next promising HP for the objective function to evaluate. Notably, learning-based SMBO is more efficient and effective than simple, non-learnable methods (Falkner et al., 2018).

However, unsupervised OD algorithms face evaluation challenges—they do not have access to (external) ground truth labels, and most of them (e.g., LOF and Isolation Forest (iForest) (Liu et al., 2008)) do not have an (internal) objective function to guide the learning either. Even for the OD algorithms with an internal objective (e.g., reconstruction loss in RAE), its value does not necessarily correlate with the actual detection performance (Ding et al., 2022). Thus, HPO for unsupervised OD is challenging and underexplored, where the key is reliable model evaluation.

Other than proposing another OD algorithm, we study this important HyperParameter Optimization for unsupervised OD problem, and introduce a systematic approach called HPOD.

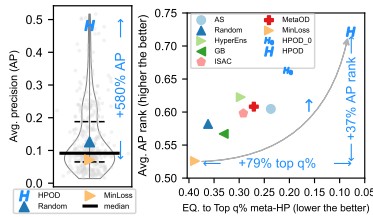 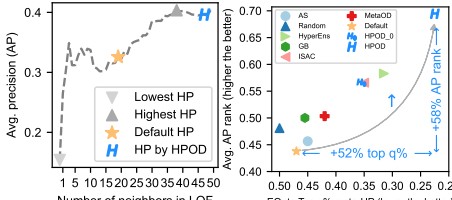 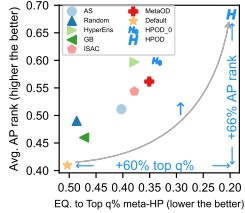

(a) (left) large perf. variation over 384 HPs for **RAE** on `Thyroid`—random&min. loss HPs are sub-par; (right) HPOD (*H*) outperforms all.

(b) (left) HP sensitivity of **LOF** on `Vowels`—the default HP is far from optimal; (right) HPOD (*H*) outperforms all baselines with +58% AP rank over random HP.

(c) Performance comparison on ensemble **iForest**; HPOD (*H*) outperforms all baselines.

Figure 1: (*a*) (left) HP sensitivity in deep RAE on `Thyroid`; (right) HPOD outperforms all baselines on a 39-dataset database (§4.2.1), with a higher avg. performance rank (y-axis) and comparable to better top q% HP settings in the meta-HP set (x-axis) (*b*) (left) HP sensitivity in LOF on `Vowels`; (right) HPOD outperforms all baselines with huge improvement (e.g., +58% norm. AP rank) over the default HP (§4.2.2) and (*c*) for iForest, HPOD is the best (e.g., +66% normalized AP rank) over the default HP (§4.2.2). See detailed experiment results in §4.

In a nutshell, HPOD leverages meta-learning to enable (originally supervised) SMBO for efficient unsupervised OD hyperparameter optimization. To overcome the infeasibility of evaluation in unsupervised OD, HPOD uses meta-learning that carries over past experience on prior datasets/tasks to more efficient learning on a new task. To that end, we build a meta-database with historical performances of a large collection of HPs on an extensive corpus of existing OD benchmark datasets, and train a *proxy performance evaluator* to evaluate HPs on a new dataset *without labels* (see §3.3). With the evaluator, HPOD can iteratively and efficiently identify promising HPs to evaluate and output the best (see §3.4). Also, we use meta-learning to further facilitate HPOD by *initializing* and *transferring knowledge* from similar historical tasks to the surrogate function of the new task (see §3.5). We remark that HPOD is strictly an HPO method other than a new detection algorithm.

**Performance**. Fig. 1a (left) shows the huge performance variation (up to 37×) for a set of 384 deep RAE HPs on `Thyroid` data, where HPOD is significantly better than expectation (i.e., random selection), as well as selection by min. reconstruction loss (MinLoss); ours is one of the top HPs. In Fig. 1a (right), we show that HPOD is significantly better than a group of diverse and competitive baselines (see Table 1) on a 39 dataset database. We also demonstrate HPOD's generality on non-deep OD algorithm LOF with both discrete and continuous HP spaces in Fig. 1b, as well as the popular iForest in Fig. 1c. For all three OD algorithms, HPOD is *statistically* better than (most) baselines, including the default HPs of widely used LOF and iForest. In fact, being an ensemble, iForest has been shown to be robust to HPs and outperform many other detectors (Emmott et al., 2015). As such, HPOD's improvement over its default HPs is remarkable.

We summarize the key contributions as follows:

- **Novel HPO Framework for Unsupervised OD**. We introduce HPOD, a meta-learning approach that capitalizes on historical OD tasks w/ labels to select effective HPs for a new task w/o labels.
- **Continuous Search and Effectiveness**. Superior to all diverse baselines in Table 1, HPOD works with both discrete *and* continuous HPs.
- **Generality**. Extensive results on 39 datasets with (*a*) deep method RAE and classical methods (*b*) LOF and (*c*) iForest show that HPOD outperforms baselines, with an avg. 37%, 58%, and 66% improvement over the minimal loss HPs of RAE and the default HPs of LOF and iForest.

We open-source HPOD and the meta-train database at `https://github.com/yzhao062/HPOD`.

## 2 Related Work

### 2.1 Hyperparameter Optimization (HPO) for OD

We can categorize the short list of HPO methods for OD into two groups. The first group of methods require a hold-out set with ground truth labels for evaluation and/or learning (Bahri et al., 2022),

Table 1: HPOD and baselines for comparison with categorization by (1st row) whether it uses meta-learning and (2nd & 3rd row) whether it supports *discrete* and *continuous* HPO. Only HPOD and HPOD_0 leverage meta-learning and support continuous HPO. See details in §4.1.

| Category | Default | Random | MinLoss | HE | GB | ISAC | AS | MetaOD | HPOD_0 | HPOD |
|---|---|---|---|---|---|---|---|---|---|---|
| **meta-learning** | ✗ | ✗ | ✗ | ✗ | ✓ | ✓ | ✓ | ✓ | ✓ | ✓ |
| **discrete HP** | ✗ | ✗ | ✓ | ✗ | ✓ | ✓ | ✓ | ✓ | ✓ | ✓ |
| **continuous HP** | ✗ | ✗ | ✓ | ✗ | ✗ | ✗ | ✗ | ✗ | ✓ | ✓ |

including AutoOD (Li et al., 2021), TODS (Lai et al., 2021), PyODDS (Li et al., 2020), and ADGym (Jiang et al., 2024), which do not apply to unsupervised OD. The second group uses the default HP, randomly picking an HP, or averaging the outputs of randomly sampled HPs (Wenzel et al., 2020); we include them as baselines (see col. 2-5 of Table 1) with empirical results in §4.2.

## 2.2 Hyperparameter Optimization and Meta-Learning

HPO gains attention due to its advantages in searching and optimizing through complex HP spaces, where learning tasks are costly (Karmaker et al., 2021). Existing methods include simple grid and random search (Bergstra and Bengio, 2012) and more efficient Sequential Model-based Bayesian Optimization (SMBO) (Jones et al., 1998). Notably, SMBO builds a cheap regression model (termed "surrogate") of the expensive objective function, and uses it to iteratively select the next promising HPs to be evaluated by the objective function (see Appx. A). We cannot directly use these supervised methods for OD. Rather, HPOD leverages meta-learning to enable efficient SMBO for *HPO for OD*.

Other than using (external) ground truth labels, a small number of studies have employed (internal) unsupervised strategies that solely use the input data and/or output outlier scores for evaluation (Marques et al., 2015; Goix, 2016a; Nguyen et al., 2017; Marques et al., 2020; Clei et al., 2022). However, a recent survey (Ma et al., 2023) shows only very few internal strategies perform better than random model selection—they are inferior to meta-learning methods (Zhao et al., 2022).

Meta-learning aims to facilitate new task learning by transferring knowledge from prior/historical tasks (Vanschoren, 2018), which has been used in warm-starting (Feurer et al., 2014, 2015) and transferring surrogate (Yogatama and Mann, 2014; Wistuba et al., 2016, 2018) in SMBO. Recently, it has also been applied to unsupervised outlier model selection (UOMS), where Zhao et al. (2021) proposed MetaOD with comparison to baselines including global best (GB), ISAC (Kadioglu et al., 2010), and ARGOSMART (AS) (Nikolic et al., 2013). We adapt these UOMS methods as baselines for *HPO for OD* (see Table 1 col. 6-9). Although these methods leverage meta-learning, they cannot handle continuous HPO. HPOD outperforms them in all experiments (see §4.2).

## 3 HPOD: Hyperparameter Optimization for Unsupervised Outlier Detection

### 3.1 Problem Statement

We consider the hyperparameter optimization (HPO) problem for unsupervised outlier detection (OD), referred to as *HPO for OD* hereafter. Given a new dataset $\mathbf{D}_{\text{test}} = (\mathbf{X}_{\text{test}}, \emptyset)$ *without any labels* and an OD algorithm $M$ with the HP space $\Lambda$, the goal is to identify a HP setting $\lambda \in \Lambda$ so that model $M_\lambda$ (i.e., detector $M$ with HP $\lambda$) achieves the highest performance[1]. HPs can be discrete and continuous, leading to an infinite number of candidate HP configurations. For instance, given $h$ hyperparameters $\lambda_1 \ldots \lambda_h$, with domains $\Lambda_1, \ldots, \Lambda_h$, the hyperparameter space $\Lambda$ of $M$ is a subset of the cross product of these domains: $\Lambda \subset \Lambda_1 \times \cdots \times \Lambda_h$. Eq. (1) presents the goal formally.

$$\operatorname*{argmax}_{\lambda \in \Lambda} \quad \text{perf}(M_\lambda, \mathbf{X}_{\text{test}}) \tag{1}$$

**Problem 1** (HPO for Unsupervised OD (HPO for OD)). Given *a new input dataset (i.e., detection task[2]) $\mathcal{D}_{\text{test}} = (\mathbf{X}_{\text{test}}, \emptyset)$ without any labels*, pick *a hyperparameter setting $\lambda \in \Lambda$ for a given detection algorithm $M$ to employ on $\mathbf{X}_{\text{test}}$ to maximize its performance.*

---

[1]In this paper, we use the area under the precision-recall curve (AUCPR, a.k.a. Average Precision or AP) as the performance metric, which can be substituted with any other metric of interest.

[2]Throughout text, we use *task* and *dataset* interchangeably.

**Algorithm 1** HPOD: Offline and Online Phases

---

**Input**: (Offline) meta-train database $\mathcal{D}_{\text{train}}$ = $\{(\mathbf{X}_i, \mathbf{y}_i)\}_{i=1}^n$, OD algorithm $M$, meta-HP set $\boldsymbol{\lambda}_{\text{meta}} = \{\boldsymbol{\lambda}_1, \ldots, \boldsymbol{\lambda}_m\} \in \Lambda$, performance evaluation perf($\cdot$); (Online) new OD dataset $\mathcal{D}_{\text{test}}$ = $(\mathbf{X}_{\text{test}}, \emptyset)$ (no labels), number of iterations $E$

**Output**: (Offline) HPOD meta-learners; (Online) the selected hyperparameter setting $\boldsymbol{\lambda}^*$ for $\mathcal{D}_{\text{test}}$

---

▶ (Offline) **Meta-train**: *Learn functions for HP performance prediction* (**§3.3**)

1: Train detector $M$ with each HP setting $\boldsymbol{\lambda}_j \in \boldsymbol{\lambda}_{\text{meta}}$ on each $\mathbf{X}_i$ of $\mathcal{D}_i \in \mathcal{D}_{\text{train}}$ to get outlier scores $\mathcal{O}_{i,j}, \forall\, i = 1 \ldots n,\ j = 1 \ldots m$

2: Evaluate each $\mathcal{O}_{i,j}$ by true labels $\mathbf{y}_i$ to get perf. matrix $\mathbf{P} \in \mathbb{R}^{n \times m}$, where $\mathbf{P}_{i,j} := \text{perf}(\mathcal{O}_{i,j}|\mathbf{y}_i)$

3: Get meta-features (MF) per task, $\mathbf{m}_i := \psi(\mathbf{X}_i)$

4: Compute internal performance measures (IPM), $\mathbf{I}_{i,j} := \phi(\mathcal{O}_{i,j})$

5: Train *proxy performance evaluator* (PPE) $f(\cdot)$ to predict the performance $\mathbf{P}_{i,j}$ from the respective $\{\text{HP } \boldsymbol{\lambda}_j, \text{MF } \mathbf{m}_i, \text{IPMs } \mathbf{I}_{i,j}\}$  ▶ §3.3.1

6: Train each *meta-surrogate function* (MSF) $t(\cdot)$ per meta-train dataset $\mathcal{T} = \{t_1, \ldots, t_n\}$ to predict the performance $\mathbf{P}_{i,j}$ from the its $\{\text{HP } \boldsymbol{\lambda}_j\}$  ▶ §3.3.2

7: **Save** MF extractor $\psi$, IPM extractor $\phi$, PPE $f$, and MSF $\mathcal{T}$

▶ (Online) **HPO on a new task**: *Iteratively identify promising HPs and output the best one* (**§3.4**)

8: Extract meta-features of the test $\mathbf{m}_{\text{test}} := \psi(\mathbf{X}_{\text{test}})$

9: Init. surrogate function $s^{(1)}$ and the evaluation set $\boldsymbol{\lambda}_{\text{eval}}$ by the meta-train and PPE  ▶ §3.4.1

10: **for** $e = 1$ to $E$ **do**  ▶ §3.4.2

11:    Transfer meta-surrogate func. $\mathcal{T}$ to surrogate $s^{(e)}$ by perf. sim. to meta-train  ▶ §3.5.2

12:    Get the promising HP to evaluate by EI on surrogates' prediction, where $\boldsymbol{\lambda}^{(e)} := \arg\max_{\boldsymbol{\lambda}_k \in \boldsymbol{\lambda}_{\text{sample}}} EI(\boldsymbol{\lambda}_k | s^{(e)})$

13:    Build $M$ with $\boldsymbol{\lambda}^{(e)}$, and get the corresponding outlier scores $\mathcal{O}_{\text{test}}^{(e)}$ and IPMs $\mathcal{I}_{\text{test}}^{(e)}$

14:    Predict performance of $\boldsymbol{\lambda}^{(e)}$ with $f(\cdot)$, i.e., $\widehat{\mathbf{P}}_{\text{test}}^{(e)} := f(\boldsymbol{\lambda}^{(e)}, \mathbf{m}_{\text{test}}, \mathcal{I}_{\text{test}}^{(e)})$

15:    Add $\boldsymbol{\lambda}^{(e)}$ to the eval. set $\boldsymbol{\lambda}_{\text{eval}} := \boldsymbol{\lambda}_{\text{eval}} \cup \boldsymbol{\lambda}^{(e)}$

16:    Update to $s^{(e+1)}$ with new pairs of information $\langle \boldsymbol{\lambda}^{(e)}, \widehat{\mathbf{P}}_{\text{test}}^{(e)} \rangle$

17: **end for**

18: **Output** $\boldsymbol{\lambda}^* \in \boldsymbol{\lambda}_{\text{eval}}$ w/ the highest predicted perf.

---

It is infeasible to evaluate an infinite number of configurations with continuous HP domain(s), and thus a key challenge is efficiently searching the space. As *HPO for OD* does not have access to ground truth labels $\mathbf{y}_{\text{test}}$, HP performance (perf.) cannot be evaluated directly.

## 3.2 **Overview of** HPOD

In HPOD, we use *meta-learning* to enable (originally supervised) Sequential Model-based Bayesian Optimization for efficient *HPO for OD*, where the key idea is to transfer useful information from historical tasks to a new test task. As such, HPOD takes as input a collection of historical tasks $\mathcal{D}_{\text{train}} = \{\mathcal{D}_1, \ldots, \mathcal{D}_n\}$, namely, a meta-train database with ground-truth labels where $\{\mathcal{D}_i = (\mathbf{X}_i, \mathbf{y}_i)\}_{i=1}^n$. Given an OD algorithm $M$ for HPO, we define a finite meta-HP set by discretizing continuous HP domains (if any) to get their cross-product, i.e., $\boldsymbol{\lambda}_{\text{meta}} = \{\boldsymbol{\lambda}_1, \ldots, \boldsymbol{\lambda}_m\} \in \Lambda$. We use $M_j$ to denote detector $M$ with the $j$-th HP setting $\boldsymbol{\lambda}_j \in \boldsymbol{\lambda}_{\text{meta}}$. HPOD uses $\mathcal{D}_{\text{train}}$ to compute:

- the historical output scores of each detector $M_j$ on each meta-train dataset $\mathcal{D}_i \in \mathcal{D}_{\text{train}}$, where $\mathcal{O}_{i,j} := M_j(\mathcal{D}_i)$ refers to the output outlier scores using the $j$-th HP setting for the points in the $i$-th meta-train dataset $\mathcal{D}_i$; and
- the historical performance matrix $\mathbf{P} \in \mathbb{R}^{n \times m}$ of each detector $M_j$, where $\mathbf{P}_{i,j} := \text{perf}(\mathcal{O}_{i,j})$ is $M_j$'s performance[1] on meta-train dataset $\mathcal{D}_i$.

HPOD consists of two phases. During **the (offline) meta-learning**, it leverages meta-train database with labels to build a *proxy performance evaluator* (PPE), which can predict HP performance of OD algorithm on a new task without labels. Also, it trains a *meta-surrogate function* (MSF) for each meta-train dataset to facilitate later HPO on a new dataset. In the **(online) HP Optimization** for a new task, HPOD uses PPE to predict its HPs' performance without using any labels, under the SMBO framework to identify promising HP settings in iteration effectively. Also, we improve the surrogate function in HPOD by transferring knowledge from *similar* meta-train datasets. An outline of HPOD is given in Algo. 1, where we show the details of offline meta-training and online *HPO for OD* on a new task in §3.3 and §3.4, respectively.

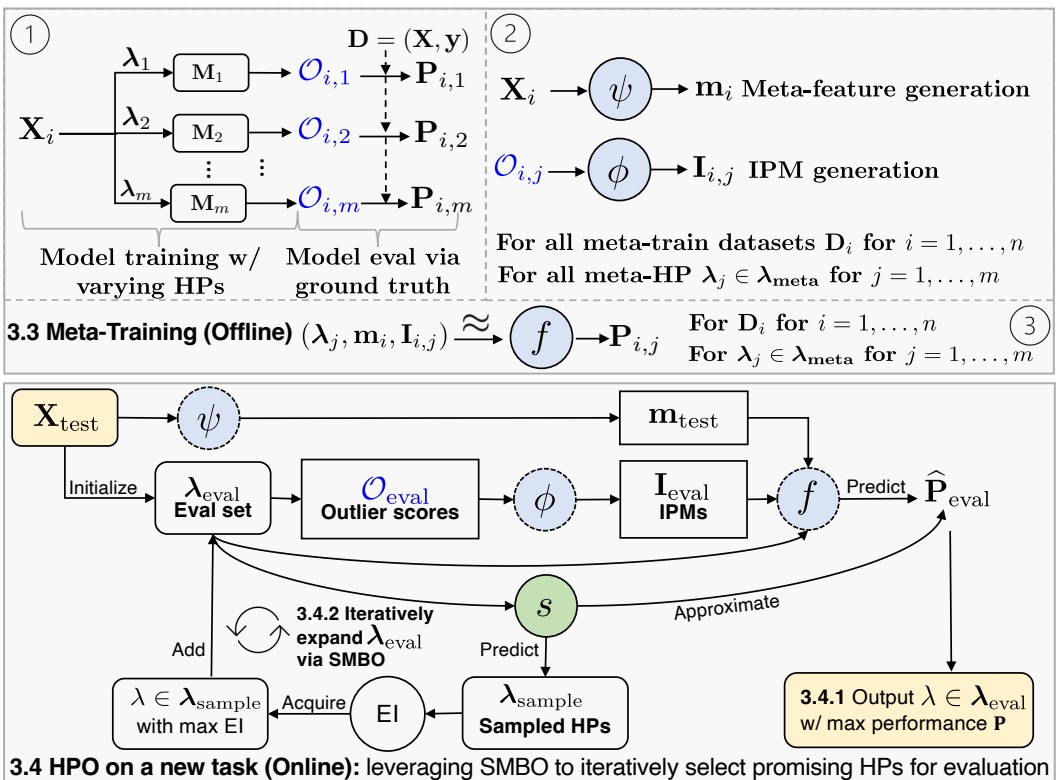

Figure 2: HPOD overview: modules we transfer from offline (meta-learning) to online (HPO) in blue; namely meta-feature extractor ($\phi$), IPM extractor ($\phi$), and proxy performance evaluator ($f(\cdot)$). In online phase, input $\mathbf{X}_{\text{test}}$ and output HP $\boldsymbol{\lambda}$ depicted in yellow, surrogate func. $s$ (in green) approximates $f(\cdot)$ for fast prediction on large set $\boldsymbol{\lambda}_{\text{sample}}$ at every SMBO iteration.

### 3.3 (Offline) Meta-training

In principle, meta-learning carries over the prior experience of historical (meta-train) tasks to do better on a new task, given the latter at least resembles some of the historical tasks. Due to the lack of ground truth labels and/or a reliable internal objective function, the key challenge in *HPO for OD* is to evaluate the performance of HP settings. Thus, the core of HPOD's meta-learning is *learning the mapping from HP settings onto ground-truth performance* by the *supervision* from the meta-train database. The first part (lines 1-7) of Algo. 1 and Fig. 2 (top) describe the core steps, and we discuss how to learn this mapping (§3.3.1) and transfer additional information for a new task (§3.3.2) in the following. Notably, *offline* meta-training is *one-time* and amortized over many test tasks.

#### 3.3.1 Proxy Performance Evaluator (PPE).

In HPOD, we learn a regressor $f(\cdot)$ across all meta-train datasets $\mathcal{D}_{\text{train}}$, named *Proxy Performance Evaluator* (PPE), that maps their {HP settings, data characteristics, additional signals} onto ground truth performances. If $f(\cdot)$ only uses HP settings as the input feature, it fails to capture the performance variation of an HP across meta-train datasets. We need additional input features to enable $f(\cdot)$ for quantifying dataset similarity, so that $f(\cdot)$ makes similar HP performance predictions on similar datasets, and vice versa.

How can we capture dataset similarity in OD? Recent work by Zhao et al. (2021) introduced specialized OD meta-features (MF) to describe general characteristics of OD datasets; e.g., number of samples, basic statistics, output statistics by certain detectors, etc. With the meta-feature extractor, both meta-train datasets and (later) the test dataset can be expressed as fixed-length vectors, and thus any similarity measure applies, e.g., Euclidean distance. To build $f(\cdot)$, we extract meta-features from each meta-train dataset as $\mathbf{M} = \{\mathbf{m}_1, \ldots, \mathbf{m}_n\} = \psi(\{\mathbf{X}_1, \ldots, \mathbf{X}_n\}) \in \mathbb{R}^{n \times d}$, where $\psi(\cdot)$ is the extraction module, and $d$ is the dimension of meta-features (see Zhao et al. (2021) for details).

Although meta-features describe general characteristics of OD datasets, their similarity does not necessarily correlate with the *actual performance.* Thus, we enrich the input features of $f(\cdot)$ with internal performance measures (IPMs) (Ma et al., 2023), which are more "performance-driven". IPMs have been proven effective in unsupervised OD model selection (Zhao et al., 2022). More specifically, IPMs are noisy/weak unsupervised signals that are solely based on the input samples and/or a given model's output (e.g., outlier scores) that can be used to compare two models (Goix, 2016b; Marques et al., 2020). In HPOD, we make the best use of these weak signals by *learning* in $f(\cdot)$ to regress the IPMs of a given HP setting (along with other signals) onto its true performance *with supervision.* To build $f(\cdot)$, we extract IPMs of each detector $M_j$ with HP setting $\lambda_j \in \lambda_{\text{meta}}$ on each meta-train dataset $\mathcal{D}_i \in \mathcal{D}_{\text{train}}$, where $\mathbf{I}_{i,j} := \phi(\mathcal{O}_{i,j})$ refers to the IPMs using the $j$-th HP for the $i$-th meta-train dataset, and $\phi(\cdot)$ is the IPM extractor. See more of IPMs in Appx. B.1.

Putting these together, we build *Proxy Performance Evaluator* $f(\cdot)$ as shown in Fig. 2 (top) to map {HP setting, meta-features, IPMs} of HP $\lambda_j \in \lambda_{\text{meta}}$ on the $i$-th meta-train dataset onto its ground truth performance, i.e., $f(\lambda_j, \mathbf{m}_i, \mathbf{I}_{i,j}) \mapsto \mathbf{P}_{i,j}$. We provide details of $f(\cdot)$ in Appx. B.2.

We want to remark that provided $\psi(\cdot)$, $\phi(\cdot)$, and the trained $f(\cdot)$ at test time, predicting the detection performance of HPs becomes possible for the new task *without* using ground-truth labels.

### 3.3.2 Meta-Surrogate Functions (MSF).
Different from $f(\cdot)$ that trains on *all* meta-train datasets and leverages *rich input features* (i.e., HPs, MFs, and IPMs) to predict HP performance, we also train $n$ *independent* regressors with *only HPs as input*, $\mathcal{T} = \{t_1, \ldots, t_n\}$. That is, for each meta-train dataset $\mathcal{D}_i \in \mathcal{D}_{\text{train}}$, we train a separate regressor $t_i(\cdot)$ that simply maps the $j$-th HP setting $\lambda_j \in \lambda_{\text{meta}}$ to its detection performance on the $i$-th meta-train dataset, i.e., $t_i(\lambda_j) \mapsto \mathbf{P}_{i,j}$.

Since these independent regressors only use HP settings as input, they can be transferred to the online HPO to improve HP performance evaluation on $\mathbf{X}_{\text{test}}$. We defer specifics to §3.4.2 & §3.5.2.

## 3.4 (Online) HPO on a New OD Task
After the meta-training phase, HPOD is ready to optimize HPs for a new dataset. In short, it outputs the HP with the highest predicted performance by $f(\cdot)$, the trained performance evaluator (§3.4.1). To explore better HPs efficiently, within time budget. HPOD leverages Sequential Model-based Optimization to iteratively select promising HPs for evaluation (§3.4.2). Lines 8-18 of Algo. 1 and Fig. 2 (bottom) show the core steps.

### 3.4.1 Hyperparameter Optimization via Proxy Performance Evaluator.
Given a new dataset $\mathcal{D}_{\text{test}}$, we can sample a set of HPs (termed as the evaluation set $\lambda_{\text{eval}} \in \Lambda$), and use the *proxy performance evaluator* $f(\cdot)$ from meta-training to predict their performance, based on which we can output the one with the highest predicted value as follows.

$$\underset{\lambda_k \in \lambda_{\text{eval}}}{\text{argmax}} \; f(\lambda_k, \mathbf{m}_{\text{test}}, \mathbf{I}_{\text{test},k}) \tag{2}$$

By setting $\lambda_{\text{eval}}$ to some *randomly sampled HPs* and plugging it into Eq. (2), we have the "version 0" of HPOD, referred as HPOD_0. However, $f(\cdot)$ needs IPMs (i.e., $\mathbf{I}_{\text{test},k}$ in Eq. (2)) as part of the input, requiring detector building at test time. Thus, we should construct $\lambda_{\text{eval}}$ carefully to ensure it captures promising HPs, where random sampling is insufficient. Thus we ask: how to efficiently identify promising HPs for model building and evaluation at test time?

### 3.4.2 Identifying Promising HPs by Sequential Model-based Optimization (SMBO).
As we briefly described in §2, SMBO can iteratively optimize an expensive objective (Hutter et al., 2011), and has been widely used in supervised model selection and HPO (Bergstra et al., 2015). Other than sampling HPs randomly, learning-based SMBO shows better efficiency in finding promising HPs to evaluate in iterations. In short, SMBO constructs a cheap regression model (called surrogate function $s(\cdot)$) and uses it for identifying the promising HPs to be evaluated by the (expensive) true objective function. It then iterates between fitting the surrogate function with newly evaluated HP information and gathering new information based on the surrogate function. We provide the pseudo-code of the *supervised* HPO by SMBO in Appx. Algo. A1, and note that it does not directly apply to *HPO for OD* as performance cannot be evaluated without ground truth labels (line 4).

We thus enable (originally *supervised*) SMBO for *unsupervised* outlier detection HPO by plugging the PPE $f(\cdot)$ from the meta-train in place of HP performance evaluation as shown in Fig. 2 (bottom).

***Surrogate Function and Initialization.*** As an approximation of the expensive objection function, surrogate function $s(\cdot)$ only takes HP settings as input, aiming for fast performance evaluation on a large collection of sampled HPs. For the new task $\mathbf{X}_{\text{test}}$ without access to true performance evaluation, HPOD lets $s(\cdot)$ learn a mapping from an HP $\boldsymbol{\lambda}_k$ to its predicted performance, i.e., $s(\boldsymbol{\lambda}_k) \mapsto f(\boldsymbol{\lambda}_k, \mathbf{m}_{\text{test}}, \mathbf{I}_{\text{test},k})$. To enable $f(\cdot)$ on $\mathbf{X}_{\text{test}}$, HPOD needs one-time computation for the corresponding meta-features as $\mathbf{m}_{\text{test}} := \psi(\mathbf{X}_{\text{test}}) \in \mathbb{R}^d$. We want to remark that $s(\cdot)$ differs from $f(\cdot)$ in two aspects. First, $s(\cdot)$ can make fast performance predictions on HPs as it only needs HPs as input, while $f(\cdot)$ is more costly since IPMs require model building. Second, $s(\cdot)$ is a regression model that can measure both *the predicted performance* of HP settings and *uncertainty (potential) around the prediction* simultaneously. A popular choice for $s(\cdot)$ is the Gaussian Process (GP)[3].

To initialize $s(\cdot)$, we train it on a small number of HPs. More specifically, we train $s(\cdot)$ with pairs of HPs and their corresponding predicted performance by $f(\cdot)$ on $\mathbf{X}_{\text{test}}$, and also initialize the evaluation set $\boldsymbol{\lambda}_{\text{eval}}$ to these HPs. Although we can randomly sample the initial HPs, we propose to set them to top-performing HPs from similar meta-train tasks (Feurer et al., 2015). Consequently, our initial $s(\cdot)$ is more accurate in predicting likely well-performing HPs on $\mathbf{X}_{\text{test}}$. We defer the details of this meta-learning-based surrogate initialization to §3.5.1.

***Iteration: Identifying Promising HPs.*** Although we can already output an HP from $\boldsymbol{\lambda}_{\text{eval}}$ with the highest predicted perf. after initialization, we aim to use $s(\cdot)$ to identify "better and better" HPs.

In each iteration, we use $s(\cdot)$ to predict the performance (denoted as $u_k := s(\boldsymbol{\lambda}_k)$) and the uncertainty around the prediction (denoted as $\sigma_k$) of sampled $\boldsymbol{\lambda}_k \in \boldsymbol{\lambda}_{\text{sample}}$ and then select the most *promising* one to be "evaluated" by $f(\cdot)$. Note that $\boldsymbol{\lambda}_{\text{sample}}$ is a HP candidate set that is randomly sampled from the full (*continuous*) HP space $\boldsymbol{\Lambda}$ (see details in Appx. C.1). Since $s(\cdot)$ can make fast predictions, $\boldsymbol{\lambda}_{\text{sample}}$'s size can be large, e.g., 10,000 as in (Hutter et al., 2011). Intuitively, we would like to evaluate the HPs with both high predicted performance $u_k$ (i.e., exploitation) and high potential/prediction uncertainty $\sigma_k$ (i.e., exploration), which is widely known as "exploitation-exploration trade-off" (Shahriari et al., 2015). Too much exploitation (i.e., always evaluating similar HPs) will fail to identify promising HPs, while too much exploration (i.e., only considering high uncertainty HPs) may lead to low-performance HPs. Also, note that the quality of identified HPs depends on the prediction accuracy of $s(\cdot)$, where we propose to transfer knowledge from meta-surrogate functions (MSF) $\mathcal{T}$ by performance similarity (see technical details in §3.5.2.)

How can we effectively balance the trade-off between exploitation and exploration in HPOD? Adapting the idea of SMBO, we use the acquisition function $a(\cdot)$ to factor in the trade-off and pick a promising HP setting based on the outputs of the surrogate function. The acquisition function quantifies the "expected utility" of HPs by balancing their predicted performance and the uncertainty. Thus, we output the most promising HP to evaluate by maximizing $a(\cdot)$:

$$\boldsymbol{\lambda} := \operatorname*{argmax}_{\boldsymbol{\lambda}_k \in \boldsymbol{\lambda}_{\text{sample}}} \quad a(s(\boldsymbol{\lambda}_k)) \tag{3}$$

One of the most prominent choices of $a(\cdot)$ is Expected Improvement (EI) (Jones et al., 1998), which is used in HPOD and can be replaced by other choices. EI has a closed-form expression under the Gaussian assumption, and the EI value of HP setting $\boldsymbol{\lambda}_k$ is shown below.

$$EI(s(\boldsymbol{\lambda}_k)) := \sigma_k \cdot [u_k \cdot \Phi(u_k) + \varphi(u_k)], \quad \text{where}$$

$$u_k = \begin{cases} \frac{u_k - \widehat{\mathbf{P}}^*_{\text{test}}}{\sigma_k} & \text{if } \sigma_k > 0 \text{ and} \end{cases} \begin{cases} 0 & \text{if } \sigma_k = 0 \end{cases} \tag{4}$$

---

[3] We use Gaussian Process (GP) (Williams and Rasmussen, 1995) here; one may use any regressor with prediction uncertainty estimation, e.g., random forests (Breiman, 2001).

In the above, $\Phi(\cdot)$ and $\varphi(\cdot)$ respectively denote the cumulative distribution and the probability density functions of a standard Normal distribution, $u_k$ and $\sigma_k$ are the predicted performance and the uncertainty around the prediction of $\lambda_k$ by the surrogate function $s(\cdot)$, and $\widehat{\mathbf{P}}^*_{\text{test}}$ is the highest predicted perf. by $f(\cdot)$ on $\lambda_{\text{eval}}$ so far. We compare EI with other selection criteria in Appx. D.7.

At iteration $e$ , we plug the surrogate $s^{(e)}(\cdot)$ into Eq. (3), which returns $\lambda^{(e)}$ to evaluate. Next, we train the OD model $M$ with $\lambda^{(e)}$ to get its scores $\mathcal{O}^{(e)}_{\text{test}}$ and IPMs $\mathcal{I}^{(e)}_{\text{test}}$, and predict its performance by $f(\cdot)$: $\widehat{\mathbf{P}}^{(e)}_{\text{test}} := f(\lambda^{(e)}, \mathbf{m}_{\text{test}}, \mathcal{I}^{(e)}_{\text{test}})$. Finally, we add $\lambda^{(e)}$ to the evaluation set $\lambda_{\text{eval}} := \lambda_{\text{eval}} \cup \lambda^{(e)}$, and update the surrogate function to $s^{(e+1)}$ with newly evaluated HP information $\langle \lambda^{(e)}, \widehat{\mathbf{P}}^{(e)}_{\text{test}} \rangle$.

As shown in Fig. 2 (bottom), HPOD alternates between (*i*) identifying the next promising HP by the surrogate function $s(\cdot)$ and (*ii*) updating $s(\cdot)$ based on newly evaluated HP and $f(\cdot)$'s outputs.
***Continuous HP search***. Recall that an outstanding property of HPOD compared to the baselines is its capability for continuous HP search (Table 1). $\lambda_{\text{sample}}$ can be *any* subset of the full HP space $\Lambda$ and not restricted to the discrete $\lambda_{\text{meta}}$.
***Time Budget***. HPOD is an *anytime* algorithm: at any time the user asks for a result, it can always output the HP with the highest predicted performance in the evaluation set $\lambda_{\text{eval}}$ at the current iteration (Eq. (2)). HPOD uses $E$ to denote the max number of iterations.

## 3.5 Details of (Online) HPO

### 3.5.1 Meta-learning-based Surrogate Initialization. 
Other than initializing on *randomly sampled* HPs, we adapt a meta-learning initialization strategy for the surrogate function $s(\cdot)$ (Feurer et al., 2015). The goal is for $s(\cdot)$ to make accurate predictions on the top-performing HPs of the test dataset, while the accuracy of the under-performing HP regions is less important. To this end, we use the meta-features (see §3.3.1) to calculate the similarity between the test dataset to each meta-train task $\mathcal{D}_i \in \mathcal{D}_{\text{train}}$, and initialize $s(\cdot)$ with the top performing HPs from *the most similar meta-train dataset*. Comparison of this scheme to random initialization can be found in Appx. D.7.

### 3.5.2 Surrogate Transfer by Performance Similarity. 
As introduced in §2.2, meta-learning can be used to improve the surrogate function $s(\cdot)$ in SMBO by transferring knowledge from meta-train datasets. In HPOD, we train $n$ independent *Meta-Surrogate Functions* (see §3.3.2) for $n$ meta-train datasets, $\mathcal{T} = \{t_1, \ldots, t_n\}$; each with the same regression function as $s(\cdot)$ (i.e., Gaussian Process). To this end, we identify the most similar meta-train dataset $\mathcal{D}_i$ in iteration $e$, and use the test surrogate $s(\cdot)$ and $\mathcal{D}_i$'s meta-surrogate $t_i$ together to predict the performance of HP $\lambda_k$, i.e.,

$$u_k := s^{(e)}(\lambda_k) + w^{(e)}_i \cdot t_i(\lambda_k) \tag{5}$$

where $w^{(e)}_i$ is the similarity between the test dataset and $\mathcal{D}_i$ measured in iteration $e$. While we could use meta-features to measure the dataset similarity, its value does not change in iterations and finds the same meta-train dataset to transfer (even for different OD algorithms).

Instead, we (re-)calculate the *performance similarity* every iteration based on the HPs in $\lambda_{\text{eval}}$ between each meta-train task and the test dataset, and *dynamically* transfer the most similar meta-train dataset's MSF. HPOD computes a rank-based similarity by weighted Kendall tau (Shieh, 1998)) between each meta-train dataset's ground truth perf. and the test dataset's predicted performance by $f(\cdot)$ on $\lambda_{\text{eval}}$ (updated in every iteration). See the effect of surrogate transfer in Appx. D.7.

## 3.6 Limitations

HPOD is designed to leverage extensive historical data to maximize its efficacy. The one-time, offline meta-training phase, though resource-intensive, is a necessary investment that significantly enhances the adaptability and performance of HPOD across diverse OD tasks. During the online phase, HPOD calculates IPMs to enable $f(\cdot)$ to predict HP performance with high precision. This step, crucial for fine-tuned optimization, does introduce additional computational demands.

Looking ahead, we will enhance its computational efficiency, e.g., refining the approximation methods for model building and exploring more efficient algorithms for performance evaluation.

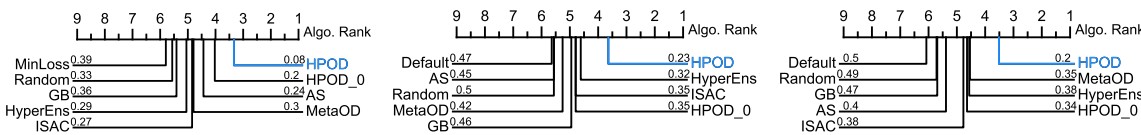

|                | (a) Results on RAE | (b) Results on LOF | (c) Results on iForest |

Figure 3: Comparison of avg. rank (lower is better) of algorithm performance across datasets on three algorithms. HPOD outperforms all w/ the lowest avg. algo. rank. The numbers on each line are the top q% value (lower is better) of the employed HP (or the avg.) by each method. HPOD shows the best performance in all three experiments.

## 4 Experiments

### 4.1 Experiment Setting

**OD Algorithms and Testbeds**. We show the results of HPO on (*a*) deep RAE in §4.2.1 and (*b*) LOF and (*c*) iForest in §4.2.2. Each OD algorithm is evaluated on a 39-dataset testbed (Appx. §D.2, Table D1). Details of each algorithm's HP spaces and the meta-HP set is provided in Appx. D.3. Experiments are conducted on an AMD 5900x@3.7GhZ, 64GB RAM workstation with RTX A6000. **Baselines**. Table 1 summarizes the baselines with categorization. We include (*i*) ***Simple methods***: **(1) Default** employs the same default/popular HP setting (only if specified in the literature) **(2) Random** choice of HPs and **(3) MinLoss** outputs the HP with the lowest internal loss (only applicable to the algorithms with an objective/loss function) and (*ii*) ***Complex methods***: **(4) HyperEnsemble (HyperEns** or **HE)** that averages the results of randomly sampled HPs (Wenzel et al., 2020) **(5) Global Best (GB)** selects the best performing HP on meta-train database on average **(6) ISAC** (Kadioglu et al., 2010) **(7) ARGOSMART (AS)** (Nikolic et al., 2013) and **(8) MetaOD** (Zhao et al., 2021). Additionally, we include **(9)** HPOD_0, a variant of HPOD that directly uses $f(\cdot)$ to choose from randomly sampled HPs (see §3.4.1). Note that the unsupervised OD model selection baselines (5)-(8) are not for *HPO for OD*, i.e., they are infeasible with continuous HP spaces. We adapt them for HPO by selecting from the discrete meta-HP set in §3.2. See baseline details in Appx. D.4. **Evaluation**. We split the meta-train/test by leave-one-out cross-validation (LOOCV). Each time we use one dataset as the input dataset for HPO, and the remaining datasets as meta-train. We run five independent trials and report the average for the baselines with randomness. We use Average Precision (AP) as the performance measure, while it can be substituted with any other measure[1]. As the raw performance like AP is not comparable across datasets with varying magnitude, we report the normalized AP rank of an HP, ranging from 1 (the best) to 0 (the worst)—thus, the higher the better. Also, we provide an additional metric called "top q%", denoting that an HP's performance has no statistical difference from the top q% HP from the meta HP-set, ranging from 0 (the best) to 1 (the worst)—thus lower the better. To compare two methods, we use the paired Wilcoxon signed rank test across all 39 datasets (significance level $p<0.05$). We give the full performance results in Appx. D.5. Also, we provide a case study on how HPOD adaptively finds better HPs in Appx. D.6. HPOD **Hyperparameters**. During meta-training, regressor $f(\cdot)$ (a LightGBM)'s, HPs are chosen via cross-validation over meta-train datasets. During HPO phase, the user time budget caps the number of online iterations $E$, as described in §3.4.2; we use a 30-minute budget for RAE and a 10-minute budget for LOF and iForest. The number of initial HPs for SMBO is set to 10.

### 4.2 Key Experiment Results (See Ablation Studies and Additional Analysis in Appx. D.7)

#### 4.2.1 Results on (Deep) Robust Autoencoder. Fig. 1a (right) and 3a show that HPOD **outperforms all baselines w.r.t. both the best avg. normalized AP rank and the top q% value**. Furthermore, HPOD is also statistically better than all baselines as shown in Table 2a, including strong meta-learning baseline MetaOD ($p=0.0398$). Its advantages can be credited to two. First, meta-learning-based HPOD leverages prior knowledge on similar historical tasks to predict HP perf. on the new dataset, whereas simple baselines like Random and MinLoss cannot. Second, only HPOD and HPOD_0 can select HPs from continuous spaces, while other meta-learning baselines are limited to finite discrete HPs as specified for the meta-train which may be too few to capture optimal HPs (especially for deep models with huge HP spaces).

| Ours | baseline | p-value | Ours | baseline | p-value |
|---|---|---|---|---|---|
| **HPOD** | AS | 0.0309 | **HPOD** | ISAC | 0.0028 |
| **HPOD** | Random | 0.0014 | **HPOD** | MetaOD | 0.0398 |
| **HPOD** | HyperEns | 0.0382 | **HPOD** | MinLoss | 0.0003 |
| **HPOD** | GB | 0.0002 | **HPOD** | HPOD_0 | 0.0201 |

(a) RAE

| Ours | baseline | p-value | Ours | baseline | p-value |
|---|---|---|---|---|---|
| **HPOD** | AS | 0.0023 | **HPOD** | ISAC | 0.0246 |
| **HPOD** | Random | 0.0001 | **HPOD** | MetaOD | 0.0088 |
| **HPOD** | **HyperEns** | 0.0607 | **HPOD** | Default | 0.0029 |
| **HPOD** | GB | 0.0017 | **HPOD** | HPOD_0 | 0.0016 |

(b) LOF

| Ours | baseline | p-value | Ours | baseline | p-value |
|---|---|---|---|---|---|
| **HPOD** | AS | 0.0055 | **HPOD** | ISAC | 0.0088 |
| **HPOD** | Random | 0.0003 | **HPOD** | MetaOD | 0.0289 |
| **HPOD** | HyperEns | 0.0484 | **HPOD** | Default | 0.0013 |
| **HPOD** | GB | 0.0027 | **HPOD** | HPOD_0 | 0.003 |

(c) iForest

Table 2: Pairwise statistical test results between HPOD and baselines by Wilcoxon signed rank test. Statistically better method shown in **bold** (both marked **bold** if no significance). (a) On RAE, HPOD is statistically better than all baselines; (b) On LOF, HPOD is statistically better than all (except HyperEnsemble (HE)), including the *default* HP setting; (c) On iForest, HPOD is statistically better than all baselines, including the *default* HP setting.

**HPO by the internal objective(s) is insufficient**. Fig. 3a shows that selecting HP by minimal loss (i.e., MinLoss) has the worst perf. for RAE, even if it can work with continuous spaces. This suggests that internal loss does not necessarily correlate with external performance. On avg., HPOD has 37% higher normalized AP rank, showing the benefit of transferring supervision via meta-learning.

**4.2.2 Results on LOF with Mixed HP Spaces and (Ensemble-based) iForest**. In addition to deep RAE, HPOD **shows generality on diverse OD algorithms**, including non-deep LOF (Fig. 1b and 3b) with mixed HP spaces (see details in Appx. Table D2b) as well as ensemble-based iForest (Fig. 1c and 3c). HPOD achieves the best performance in both with the best norm. AP rank and top q%.

HPOD **is statistically better than the default HPs of LOF ($p$=0.0029) and iForest ($p$=0.0013)** (see Table 2b and 2c). More specifically, we find that HPOD provides +58% and +66% performance (i.e., normalized AP rank) improvement over using the default HPs of LOF (Fig. 1b (right)) and iForest (Fig. 1c). In fact, note that the default HPs rank the lowest for both LOF (Fig. 3b) and iForest (Fig. 3c), justifying the importance of HPO methods in unsupervised OD.

**HyperEns that averages outlier scores from randomly sampled HPs yield reasonable performance**, which agrees with the observations in the literature (Ding et al., 2022). However, it has a higher inference cost as it needs to save and use all base models, not ideal for time-critical applications. Using a single model with the selected HPs by HPOD offers better accuracy and efficiency.

## 5 Conclusion

We introduce (to our knowledge) the *first systematic continuous hyperparameter optimization (HPO) approach for unsupervised outlier detection (OD)*. The proposed HPOD is a meta-learner, and builds on an extensive pool of *existing* OD benchmark datasets based on which it trains a performance predictor (offline). Given a new task without labels (online), it capitalizes on the performance predictor to enable (originally supervised) sequential model-based optimization for identifying promising HPs iteratively. Notably, HPOD stands out from all prior work on *HPO for OD* in being capable of handling both discrete *and* continuous HPs. Extensive experiments on three (including both deep and shallow) OD algorithms show its generality, where it significantly outperforms a diverse set of baselines. Future work will consider joint algorithm selection and continuous hyperparameter optimization for unsupervised outlier detection.

## 6 Broader Impact Statement

Automating HP-tuning for OD via HPOD offers significant benefits, especially for practitioners struggling to select suitable HPs for their unlabeled tasks. In practice, many users resort to pre-set default HP values specified in OD software packages, which we have shown significantly underperform compared to the hyperparameters as optimized by HPOD.

OD is utilized in various sectors including security and surveillance, finance, manufacturing, and healthcare. These include adversarial applications such as fraud detection, wherein adversaries may adjust to evade optimally tuned detectors. This adversarial dynamic continues the "cat-and-mouse" cycle, driving the development of new algorithms to counteract evasion. HPOD can HP-tune any detector, and we expect it will continue to be useful for future detectors in unsupervised settings.

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

## Supplementary Material for HPOD

*Details on algorithm design and experiments.*

## A Details on Supervised SMBO

Algorithm A1 shows the pseudo-code of classical SMBO for HPO. In each iteration, the surrogate function $s(\cdot)$ predicts the performance and uncertainty of a group of sampled HP settings, where the acquisition function $a(\cdot)$ selects the best one to be evaluated next by the objective function $\mathcal{L}(\cdot)$. With the newly evaluated pair of HP settings and the objective value, the surrogate function is updated to be more accurate in iteration.

---

**Algorithm A1** SMBO for Supervised Hyperparameter Optimization

---

**Input**: learning algorithm $M$, surrogate function $s(\cdot)$, input task $\mathbf{D}_{\text{test}} = \{\mathbf{X}_{\text{test}}, \mathbf{y}_{\text{test}}\}$, objective function $\mathcal{L}(\cdot)$, number of iterations $E$

**Output**: selected hyperparameter setting $\boldsymbol{\lambda}^*$ for $\mathbf{D}_{\text{test}}$

---

1: Initialize surrogate function $s^{(1)}$
2: **for** $e = 1$ to $E$ **do**
3:     $\boldsymbol{\lambda}^{(e)} := \text{argmax}_{\boldsymbol{\lambda} \in \Lambda} EI(\mathbf{X}_{\text{test}}), \boldsymbol{\lambda}|M, s^{(e)}$
4:     $\mathcal{L}^{(e)} := \text{evaluate } \mathcal{L}(M_{\boldsymbol{\lambda}^{(e)}}, \mathbf{X}_{\text{test}}, \mathbf{y}_{\text{test}})$              ▶ infeasible for *HPO for OD*
5:     Update to $s^{(e+1)}$ with new information $\langle \boldsymbol{\lambda}^{(e)}, \mathcal{L}^{(e)} \rangle$
6: **end for**
7: **Output** $\boldsymbol{\lambda}^* \in \boldsymbol{\lambda}^{(1)}, \ldots, \boldsymbol{\lambda}^{(e)}$ with the highest evaluated objective function values

---

Clearly, the classical SMBO does not apply to *HPO for OD* directly since the objective function $\mathcal{L}(\cdot)$ cannot be evaluated without ground truth labels (line 4). The proposed HPOD uses meta-learning to train a regressor $f(\cdot)$ to predict the performance of an HP on the new dataset without any labels (§3.3), and thus enables (originally supervised) SMBO for *HPO for OD* (§3.4).

## B (Offline) Meta-training Details

### B.1 Internal Performance Measures (IPMs)

As described in §3.3.1, IPMs are used as part of the input features of the *proxy performance evaluator* $f(\cdot)$. In HPOD, we use three consensus-based IPMs (i.e., MC, SELECT, and HITS) which carry useful and noisy signals in unsupervised OD model selection (Ma et al., 2023). in short, consensus-based IPMs consider the resemblance to the overall consensus of outlier scores as a sign of a better (performance of) model. Thus, a group of models is needed to compute these IPMs for resemblance measure.

In (Ma et al., 2023), they use all models in $\mathcal{M}$ for building IPMs (i.e., $\mathcal{M} = \{M_1, \ldots, M_m\}$ by pairing detector $M$ with each HP in meta-HP set $\boldsymbol{\lambda}_{\text{meta}} = \{\boldsymbol{\lambda}_1, \ldots, \boldsymbol{\lambda}_m\} \in \Lambda$), leading to high cost in generating outlier scores and then IPMs. To reduce the cost, we instead identify a small subset of representative models $\mathcal{M}_A \in \mathcal{M}$ called the *anchor* set (i.e., $|\mathcal{M}_A| \ll |\mathcal{M}|$), for calculating IPMs. That is, we generate the IPMs of a model with regard to its consensus to $\mathcal{M}_A$ rather than $\mathcal{M}$, for both the meta-train database and the input dataset. The construction of the anchor set can be done by cross-validation in a forward selection way.

### B.2 Building Proxy Performance Evaluator (PPE)

As outlined in §3.3.1, we build *Proxy Performance Evaluator* $f(\cdot)$ to map {HP setting, meta-features, IPMs} of HP $\boldsymbol{\lambda}_j \in \boldsymbol{\lambda}_{\text{meta}}$ on the $i$-th meta-train dataset onto its ground truth performance, i.e., $f(\boldsymbol{\lambda}_j, \mathbf{m}_i, \mathbf{I}_{i,j}) \mapsto \mathbf{P}_{i,j}$. Given we have $n$ meta-train datasets and the meta-HP set with $|\boldsymbol{\lambda}_{\text{meta}}| = m$ HP settings, $f(\cdot)$ is trained on $mn$ samples by pairing $\boldsymbol{\lambda}_{\text{meta}}$ with meta-train datasets.

To construct the training samples of $f(\cdot)$, we first extract meta-features from each meta-train dataset as $\mathbf{M} = \{\mathbf{m}_1, \ldots, \mathbf{m}_n\} = \psi(\{\mathbf{X}_1, \ldots, \mathbf{X}_n\}) \in \mathbb{R}^{n \times d}$, where $\psi(\cdot)$ is the extraction module, and $d$ is the dimension of meta-features.

We also need to extract IPMs of each detector $M_j$ with HP setting $\boldsymbol{\lambda}_j \in \boldsymbol{\lambda}_{\mathbf{meta}}$ on each meta-train dataset $\mathcal{D}_i \in \boldsymbol{\mathcal{D}}_{\text{train}}$, where $\mathbf{I}_{i,j} := \phi(\mathcal{O}_{i,j})$ refers to the IPMs using the $j$-th HP setting the $i$-th meta-train dataset and $\phi$ is the extractor.

Putting these together, we train $f(\cdot)$ with $(m \cdot n)$ samples. In implementation we use LightGBM (Ke et al., 2017) for $f(\cdot)$, while it is flexible to choose any other.

### B.3 Meta-surrogate Functions (MSF)

As described in §3.3.2, we also train $n$ *independent* regressors with *only HPs as input*, $\boldsymbol{\mathcal{T}} = \{t_1, \ldots, t_n\}$. That is, for each meta-train dataset $\mathcal{D}_i \in \boldsymbol{\mathcal{D}}_{\text{train}}$, we train a regressor $t_i(\cdot)$ that simply maps the $j$-th HP setting $\boldsymbol{\lambda}_j \in \boldsymbol{\lambda}_{\mathbf{meta}}$ to its detection performance on the $i$-th meta-train dataset, i.e., $t_i(\boldsymbol{\lambda}_j) \mapsto \mathbf{P}_{i,j}$. Thus, $t_i(\cdot)$ only trains on the $m$ HP settings' performance on the $i$-th meta-train dataset. In implementation, we use Gaussian Process (GP) (Williams and Rasmussen, 1995) for MSF [3], and we suggest using the same regressor as the surrogate $s(\cdot)$ in §3.4 for easy knowledge transfer in §3.5.2.

## C (Online) Model Selection Details

### C.1 Sampling Range

Given the PPE, $f(\cdot)$, is trained on the meta-HP set $\boldsymbol{\lambda}_{\mathbf{meta}}$ of the meta-train database, it is more accurate in predicting the HPs from a similar range for the new dataset. Thus, HPOD samples HPs within the range of $\boldsymbol{\lambda}_{\mathbf{meta}}$ in SMBO (see §3.4.2). For instance, given the meta-HP set of iForest shown in Appx. Table D2c, we sample HPs in range of: (*i*) n_estimators in $[10, 150]$ (*ii*) max_samples in $[0.1, 0.9]$ and (*iii*) max_features in $[0.2, 0.8]$ for $\boldsymbol{\lambda}_{\mathbf{sample}}$. We provide more details on the fast simulation of sampling in Appx. D.3.

## D Additional Experiment Settings and Results

### D.1 Code and Reproducibility

We foster future research by fully releasing the code and the testbed at repo: `https://github.com/yzhao062/HPOD`.

### D.2 Datasets

In Table D1, we describe the details of the 39 benchmark datasets used in the experiments—it is composed by 18 datasets from DAMI library (Campos et al., 2016) and 21 datasets from ODDS library (Rayana, 2016).

Note that HPOD can be extended with more benchmark datasets, and we expect its performance can be further improved.

### D.3 OD Algorithms and Hyperparameter Spaces

We demonstrate the HPOD effectiveness on three OD algorithms, namely RAE, LOF, and iForest. For RAE, we adapt the author's code with seven key HPs. For LOF and iForest, we use the implementation from Python Outlier Detection (PyOD) library. For fast simulation, we also precompute the outlier scores and IPMs for the inner-HP set (denoted as $\boldsymbol{\lambda}_{\mathbf{inner}}$), which is within the range of the meta-HP set and serving as additional HPs sampled from continuous HP spaces. In the experiment, HPOD sets $\boldsymbol{\lambda}_{\mathbf{sample}} = \boldsymbol{\lambda}_{\mathbf{meta}} \cup \boldsymbol{\lambda}_{\mathbf{inner}}$, and uses $s(\cdot)$ to score all the HPs in $\boldsymbol{\lambda}_{\mathbf{sample}}$ that are not yet evaluated by $f(\cdot)$ yet (see §3.4), thus simulating the advantage of sampling from larger "continuous" HP spaces. Table D2 and the code show details of HP spaces, the meta-HP set, and the inner-HP set.

Table D1: Testbed composed of 18 datasets from DAMI library and 21 datasets from ODDS library.

| Dataset | Source | #Samples | #Dims | %Outlier |
|---|---|---|---|---|
| DAMI_ALOI | DAMI | 49534 | 27 | 3.04 |
| DAMI_Annthyroid | DAMI | 7129 | 21 | 7.49 |
| DAMI_Arrhythmia | DAMI | 450 | 259 | 45.78 |
| DAMI_Cardiotocography | DAMI | 2114 | 21 | 22.04 |
| DAMI_Glass | DAMI | 214 | 7 | 4.21 |
| DAMI_HeartDisease | DAMI | 270 | 13 | 44.44 |
| DAMI_InternetAds | DAMI | 1966 | 1555 | 18.72 |
| DAMI_PageBlocks | DAMI | 5393 | 10 | 9.46 |
| DAMI_PenDigits | DAMI | 9868 | 16 | 0.2 |
| DAMI_Pima | DAMI | 768 | 7 | 34.9 |
| DAMI_Shuttle | DAMI | 1013 | 9 | 1.28 |
| DAMI_SpamBase | DAMI | 4207 | 57 | 39.91 |
| DAMI_Stamps | DAMI | 340 | 9 | 9.12 |
| DAMI_Waveform | DAMI | 3443 | 21 | 2.9 |
| DAMI_WBC | DAMI | 223 | 9 | 4.48 |
| DAMI_WDBC | DAMI | 367 | 30 | 2.72 |
| DAMI_Wilt | DAMI | 4819 | 5 | 5.33 |
| DAMI_WPBC | DAMI | 198 | 33 | 23.74 |
| ODDS_annthyroid | ODDS | 7200 | 6 | 7.42 |
| ODDS_arrhythmia | ODDS | 452 | 274 | 14.6 |
| ODDS_breastw | ODDS | 683 | 9 | 34.99 |
| ODDS_glass | ODDS | 214 | 9 | 4.21 |
| ODDS_ionosphere | ODDS | 351 | 33 | 35.9 |
| ODDS_letter | ODDS | 1600 | 32 | 6.25 |
| ODDS_lympho | ODDS | 148 | 18 | 4.05 |
| ODDS_mammography | ODDS | 11183 | 6 | 2.32 |
| ODDS_mnist | ODDS | 7603 | 100 | 9.21 |
| ODDS_musk | ODDS | 3062 | 166 | 3.17 |
| ODDS_optdigits | ODDS | 5216 | 64 | 2.88 |
| ODDS_pendigits | ODDS | 6870 | 16 | 2.27 |
| ODDS_pima | ODDS | 768 | 8 | 34.9 |
| ODDS_satellite | ODDS | 6435 | 36 | 31.64 |
| ODDS_satimage-2 | ODDS | 5803 | 36 | 1.22 |
| ODDS_speech | ODDS | 3686 | 400 | 1.65 |
| ODDS_thyroid | ODDS | 3772 | 6 | 2.47 |
| ODDS_vertebral | ODDS | 240 | 6 | 12.5 |
| ODDS_vowels | ODDS | 1456 | 12 | 3.43 |
| ODDS_wbc | ODDS | 378 | 30 | 5.56 |
| ODDS_wine | ODDS | 129 | 13 | 7.75 |

### D.4 Baselines

We provide the details of baselines presented in Table 1 and §4.1, namely simple methods and complex methods.

*Simple methods*:

(1) **Default** always employs the same default/popular HP setting of the underlying OD algorithm (only applicable to the algorithms with recommended HPs).

(2) **Random** denotes selecting HPs randomly.

(3) **MinLoss** outputs the HP with the lowest internal loss (only applicable to the algorithms with an internal objective/loss) from a group of random samples HPs.

*Complex methods*:

(4) **Hyperensemble (HyperEns or HE)** that averages the outlier scores of randomly sampled HPs (Wenzel et al., 2020). Strictly speaking, HE is not an HPO method.

(5) **Global Best (GB)** selects the best performing HP on meta-train database on average.

Table D2: Key HPs optimized by HPOD, and the meta-HP set and the inner-HP set used in this study.

| | HP Name | Type | Meta-HP Set | Inner-HP Set |
|---|---|---|---|---|
| 1 | # EncodeLayers | int (continuous) | {2,4} | {2,4} |
| 2 | Lambda | float (continuous) | {5e-5, 5e-3, 5e-1} | {1e-4, 1e-3, 1e-1} |
| 3 | Learning Rate | float (continuous) | {1e-3, 1e-2} | {1e-3, 1e-2} |
| 4 | # Inner Epochs | int (continuous) | {20, 50} | {30, 40} |
| 5 | # Outer Epochs | int (continuous) | {20, 50} | {30, 40} |
| 6 | Shrinkage Decay | int (continuous) | {2,4} | {2,4} |
| 7 | Dropout | float (continuous) | {0, 0.1, 0.3, 0.5} | {0, 0.1, 0.2, 0.4} |

(a) Key HPs optimized by HPOD for RAE. Both meta-HP set and inner-HP set include $2 \times 3 \times 2 \times 2 \times 2 \times 2 \times 3$=388 HP settings.

| | HP Name | Type | Meta-HP Set | Inner-HP Set |
|---|---|---|---|---|
| 1 | n_neighbors | int (continuous) | {1,3,5,. . .,80} | {2,4,6,. . .,81} |
| 2 | distance metric | str (categorical) | {'chebyshev', 'minkowski', 'cosine', 'euclidean', 'manhattan'} | Same |

(b) Key HPs optimized by HPOD for LOF. Both meta-HP set and inner-HP set include $40 \times 5$=200 HP settings.

| | HP Name | Type | Meta-HP Set | Inner-HP Set |
|---|---|---|---|---|
| 1 | n_estimators | int (continuous) | {10,20,30,40,50,75,100,150} | {10,20,30,40,50,75,100,150} |
| 2 | max_samples | float (continuous) | {0.1, 0.2, . . . , 0.9} | {0.1, 0.2, . . . , 0.9} |
| 3 | max_features | float (continuous) | {0.2, 0.4, 0.6, 0.8} | {0.3, 0.5, 0.7, 0.75} |

(c) Key HPs optimized by HPOD for iForest. Both meta-HP set and inner-HP set include $8 \times 9 \times 4$=288 HP settings.

(6) **ISAC** (Kadioglu et al., 2010) first groups meta-train datasets into clusters, and assigns the best performing HP in the meta-HP set to each cluster. During the online HPO phase, it first assigns the new dataset to one of the clusters and uses the group-based HP for the new dataset.

(7) **ARGOSMART (AS)** (Nikolic et al., 2013) identifies the most similar meta-train dataset of the new task, and outputs the best performing HP on the meta-task for the new task.

(8) **MetaOD** (Zhao et al., 2021) uses matrix factorization to capture the dataset similarity and HPs' performance similarity, which is the SOTA unsupervised outlier model selection method.

Additionally, we include (9) HPOD_0, a variant of HPOD that directly uses $f(\cdot)$ to choose from randomly sampled HPs (see §3.4.1). Note that these unsupervised OD model selection baselines (5-8) are not original for HPO for OD, and could not work with continuous HP spaces. We adapt them for HPO by selecting an HP from the meta-HP set described in §3.2.

## D.5 Full Performance Results

In addition to the avg. rank plot in Fig. 3, we provide the full performance of RAE in Table D3, as well as the results for LOF and iForest in Table D4 and D5, respectively.

Table D3: Method evaluation on RAE (normalized AP rank). The best performing method is highlighted in bold. The algo. rank is provided in parenthesis (lower ranks denote better performance). HPOD achieves the best performance among all baselines.

| datasets | Random | GB | ISAC | AS | HyperEns | MetaOD | MinLoss | HPOD_0 | HPOD |
|---|---|---|---|---|---|---|---|---|---|
| DAMI_ALOI | 0.8234 (3) | 0.6445 (4) | 0.6445 (4) | 0.0612 (8) | 0.6286 (6) | 0.0612 (8) | 0.3747 (7) | 0.9206 (2) | **0.974 (1)** |
| DAMI_Annthyroid | 0.5506 (5) | 0.0404 (9) | 0.4336 (6) | 0.5508 (4) | 0.2499 (8) | 0.8659 (2) | 0.5651 (3) | 0.3211 (7) | **0.9883 (1)** |
| DAMI_Arrhythmia | 0.4688 (9) | 0.8112 (5) | 0.7435 (6) | **0.9987 (1)** | 0.4836 (8) | 0.9036 (3) | 0.7013 (7) | 0.9451 (2) | 0.9036 (3) |
| DAMI_Cardiotocography | 0.6519 (5) | 0.5951 (6) | 0.819 (3) | 0.1432 (9) | 0.6727 (4) | 0.2786 (8) | 0.5544 (7) | 0.949 (2) | **0.9688 (1)** |
| DAMI_Glass | 0.6403 (3) | 0.5781 (6) | 0.4036 (7) | 0.1823 (9) | **0.7065 (1)** | 0.6094 (5) | 0.2651 (8) | 0.6549 (2) | 0.6354 (4) |
| DAMI_HeartDisease | 0.5156 (3) | 0.3958 (6) | 0.5156 (2) | 0.056 (8) | **0.5662 (1)** | 0.1901 (7) | 0.4128 (5) | 0.4487 (4) | 0.056 (8) |
| DAMI_InternetAds | 0.6623 (2) | 0.5156 (7) | 0.5156 (7) | 0.5221 (6) | **0.6743 (1)** | 0.5404 (5) | 0.3419 (9) | 0.6549 (3) | 0.651 (4) |
| DAMI_PageBlocks | 0.7584 (5) | 0.7591 (3) | 0.7591 (3) | **0.9779 (1)** | 0.6078 (7) | 0.3047 (8) | 0.7544 (6) | 0.3047 (8) | 0.9505 (2) |
| DAMI_PenDigits | 0.4909 (4) | 0.0391 (9) | 0.1289 (7) | 0.4909 (5) | **0.8 (1)** | 0.4909 (5) | 0.5247 (3) | 0.619 (2) | 0.1289 (7) |
| DAMI_Pima | 0.4117 (8) | 0.8698 (3) | 0.8698 (3) | 0.9779 (1) | 0.7792 (5) | 0.4115 (9) | 0.6865 (6) | 0.5247 (7) | **0.9779 (1)** |
| DAMI_Shuttle | 0.3195 (2) | 0.319 (3) | 0.319 (3) | 0.319 (3) | **0.3849 (1)** | 0.319 (3) | 0.2464 (7) | 0.0784 (8) | 0.056 (9) |
| DAMI_SpamBase | 0.3208 (6) | 0.1276 (8) | 0.1276 (8) | **0.9323 (1)** | 0.4618 (4) | 0.7214 (2) | 0.5242 (3) | 0.194 (7) | 0.3364 (5) |
| DAMI_Stamps | 0.6727 (5) | 0.6302 (6) | 0.1133 (9) | 0.5924 (7) | 0.7777 (4) | **0.9844 (1)** | 0.5318 (8) | 0.9026 (3) | 0.9714 (2) |
| DAMI_Waveform | 0.687 (8) | 0.7018 (7) | 0.8242 (5) | 0.8932 (2) | 0.5257 (9) | 0.8932 (2) | 0.7977 (6) | **0.9513 (1)** | 0.8932 (2) |
| DAMI_WBC | 0.3403 (4) | 0.3398 (5) | **0.8242 (1)** | 0.1628 (8) | 0.6966 (2) | 0.6328 (3) | 0.0648 (9) | 0.2628 (6) | 0.2435 (7) |
| DAMI_WDBC | 0.5896 (5) | 0.3542 (7) | 0.0378 (9) | 0.9115 (2) | 0.5849 (6) | 0.3542 (7) | 0.7099 (4) | 0.7219 (3) | **0.9935 (1)** |
| DAMI_Wilt | 0.5013 (8) | 0.5013 (1) | **0.5013 (1)** | 0.5013 (1) | 0.0026 (9) | 0.5013 (1) | 0.5013 (1) | **0.5013 (1)** | 0.5013 (1) |
| DAMI_WPBC | 0.5039 (5) | 0.2943 (8) | 0.5846 (4) | 0.737 (2) | 0.2899 (9) | 0.6354 (3) | **0.7562 (1)** | 0.4445 (7) | 0.5039 (5) |
| ODDS_annthyroid | 0.6351 (7) | 0.3112 (8) | **0.9922 (1)** | 0.9701 (2) | 0.8379 (5) | 0.8307 (6) | 0.131 (9) | 0.8703 (4) | 0.9312 (3) |
| ODDS_arrhythmia | 0.4753 (9) | 0.6432 (7) | 0.9714 (2) | 0.9596 (4) | 0.5143 (8) | 0.7344 (5) | 0.6883 (6) | 0.969 (3) | **0.9844 (1)** |
| ODDS_breastw | 0.139 (6) | 0.0716 (9) | 0.6719 (2) | 0.138 (7) | **0.9818 (1)** | 0.138 (7) | 0.2052 (5) | 0.6719 (2) | 0.6714 (4) |
| ODDS_glass | 0.4584 (6) | 0.6589 (3) | 0.6693 (2) | 0.375 (8) | 0.5475 (4) | **0.7865 (1)** | 0.1318 (9) | 0.3802 (7) | 0.5312 (5) |
| ODDS_ionosphere | 0.7039 (9) | 0.8047 (6) | 0.8216 (5) | **0.9844 (1)** | 0.7143 (8) | 0.7969 (7) | 0.8307 (4) | 0.9096 (3) | 0.9351 (2) |
| ODDS_letter | 0.5013 (3) | 0.0872 (7) | 0.0182 (8) | 0.0182 (8) | 0.5273 (2) | **0.918 (1)** | 0.4492 (4) | 0.1643 (6) | 0.2591 (5) |
| ODDS_lympho | 0.7506 (8) | 0.7695 (7) | 0.8802 (5) | 0.9583 (3) | 0.6545 (9) | 0.974 (2) | 0.7805 (6) | 0.9177 (4) | **1 (1)** |
| ODDS_mammography | 0.4844 (6) | 0.3607 (9) | 0.4844 (7) | 0.3841 (8) | 0.5896 (4) | **0.9349 (1)** | 0.7328 (3) | 0.531 (5) | 0.7865 (2) |
| ODDS_mnist | 0.6052 (6) | **0.9818 (1)** | 0.7786 (3) | 0.2513 (9) | 0.5496 (7) | 0.6406 (5) | 0.4677 (8) | 0.7279 (4) | 0.9401 (2) |
| ODDS_musk | 0.5584 (8) | 0.901 (2) | 0.6289 (6) | **0.9857 (1)** | 0.5922 (7) | 0.7018 (5) | 0.1414 (9) | 0.8112 (4) | 0.8516 (3) |
| ODDS_optdigits | 0.5286 (5) | 0.1406 (8) | 0.5977 (3) | 0.2122 (7) | 0.6509 (2) | 0.5977 (3) | **0.794 (1)** | 0.2721 (6) | 0.0755 (9) |
| ODDS_pendigits | 0.626 (3) | 0.5169 (6) | 0.5299 (5) | 0.1445 (8) | **0.8119 (1)** | 0.0924 (9) | 0.318 (7) | 0.7096 (2) | 0.613 (4) |
| ODDS_pima | 0.713 (4) | 0.7135 (3) | 0.569 (7) | 0.569 (7) | 0.6312 (6) | 0.569 (7) | 0.6779 (5) | 0.9247 (2) | **0.9987 (1)** |
| ODDS_satellite | 0.8714 (4) | 0.8503 (6) | 0.8685 (5) | **0.9909 (1)** | 0.7813 (7) | 0.2865 (9) | 0.7471 (8) | 0.9471 (2) | 0.9323 (3) |
| ODDS_satimage-2 | 0.9143 (3) | 0.8333 (6) | 0.8737 (5) | **0.987 (1)** | 0.7927 (7) | 0.3997 (9) | 0.481 (8) | 0.9063 (4) | 0.9506 (2) |
| ODDS_speech | 0.6299 (7) | 0.9714 (3) | 0.4154 (9) | 0.9792 (2) | 0.4842 (8) | 0.8177 (4) | 0.6773 (6) | 0.699 (5) | **0.9909 (1)** |
| ODDS_thyroid | 0.6468 (8) | 0.9245 (4) | 0.8633 (6) | 0.9453 (2) | **0.9647 (1)** | 0.7305 (7) | 0.2867 (9) | 0.9164 (5) | 0.9247 (3) |
| ODDS_vertebral | 0.6974 (8) | 0.6979 (3) | 0.6979 (3) | 0.6979 (3) | 0.4649 (9) | 0.6979 (3) | 0.7487 (2) | **0.8133 (1)** | 0.6979 (3) |
| ODDS_vowels | 0.7506 (7) | 0.9349 (4) | 0.4648 (9) | **0.9974 (1)** | 0.6987 (8) | 0.8581 (6) | 0.869 (5) | 0.9846 (2) | 0.9818 (3) |
| ODDS_wbc | 0.6195 (6) | **0.9401 (1)** | 0.8815 (3) | 0.5443 (7) | 0.6566 (5) | 0.5443 (7) | 0.456 (9) | 0.9117 (2) | 0.8727 (4) |
| ODDS_wine | 0.4818 (4) | 0.4818 (5) | 0.4818 (5) | 0.4818 (5) | 0.9403 (2) | **0.9714 (1)** | 0.5797 (3) | 0.387 (9) | 0.4818 (5) |
| Average | 0.5821 (7) | 0.567 (8) | 0.5981 (6) | 0.6047 (5) | 0.6226 (3) | 0.6082 (4) | 0.5258 (9) | 0.6622 (2) | **0.7216 (1)** |
| STD | 0.1580 | 0.2884 | 0.2630 | 0.3478 | 0.1937 | 0.2647 | 0.2261 | 0.2718 | 0.3103 |
| Avg. Rank | 5.5641 | 5.4103 | 4.8462 | 4.4359 | 5.0513 | 4.7949 | 5.7949 | 4.0256 | **3.3333** |

Table D4: Method evaluation on LOF (normalized AP rank). The best performing method is highlighted in bold. The algo. rank is provided in parenthesis (lower ranks denote better performance). HPOD achieves the best performance among all baselines.

| datasets | Random | GB | ISAC | AS | HyperEns | MetaOD | Default | HPOD_0 | HPOD |
|---|---|---|---|---|---|---|---|---|---|
| DAMI_ALOI | 0.6418 (5) | 0.6825 (4) | 0.625 (6) | **0.95 (1)** | 0.005 (9) | 0.1319 (8) | 0.905 (2) | 0.8975 (3) | 0.3433 (7) |
| DAMI_Annthyroid | 0.5224 (7) | 0.6175 (5) | 0.565 (6) | 0.0275 (8) | 0.8836 (3) | 0.9236 (2) | **0.97 (1)** | 0.0125 (9) | 0.6841 (4) |
| DAMI_Arrhythmia | 0.3134 (8) | 0.4125 (5) | 0.4 (6) | **0.98 (1)** | 0.8557 (3) | 0.2188 (9) | 0.335 (7) | 0.92 (2) | 0.7761 (4) |
| DAMI_Cardiotocography | 0.4776 (4) | 0.31 (6) | 0.745 (3) | 0.19 (8) | **1 (1)** | 0.4583 (5) | 0.165 (9) | 0.2225 (7) | 0.985 (2) |
| DAMI_Glass | 0.5721 (3) | 0.3975 (5) | 0.725 (2) | 0.02 (9) | 0.2647 (7) | 0.1684 (8) | **0.95 (1)** | 0.3525 (6) | 0.5498 (4) |
| DAMI_HeartDisease | 0.5473 (6) | 0.7625 (3) | 0.4825 (7) | 0.035 (9) | **1 (1)** | 0.6319 (4) | 0.32 (8) | 0.555 (5) | 0.8507 (2) |
| DAMI_InternetAds | 0.5174 (5) | 0.5675 (3) | **1 (1)** | 0.39 (7) | 0.4825 (6) | 0.9132 (2) | 0.02 (9) | 0.2775 (8) | 0.5572 (4) |
| DAMI_PageBlocks | 0.4129 (6) | 0.4425 (5) | **0.985 (1)** | 0.235 (8) | 0.61 (3) | 0.5903 (4) | 0.24 (7) | 0.155 (9) | 0.8507 (2) |
| DAMI_PenDigits | 0.3333 (7) | 0.3475 (6) | 0.69 (4) | 0.4 (5) | 0.1333 (8) | 0.8924 (3) | 0.9 (1) | 0.075 (9) | **0.9 (1)** |
| DAMI_Pima | 0.3184 (8) | 0.535 (6) | **0.9975 (1)** | 0.47 (7) | 0.99 (2) | 0.783 (4) | 0.02 (9) | 0.59 (5) | 0.9125 (3) |
| DAMI_Shuttle | 0.5174 (6) | 0.2725 (8) | 0.52 (5) | 0.6925 (3) | 0.0229 (9) | 0.309 (7) | **0.895 (1)** | 0.7125 (2) | 0.6925 (3) |
| DAMI_SpamBase | 0.403 (5) | 0.4075 (4) | 0.21 (6) | 0.17 (8) | **1 (1)** | 0.1823 (7) | 0.155 (9) | 0.715 (3) | 0.8259 (2) |
| DAMI_Stamps | 0.408 (4) | 0.4225 (3) | 0.8 (2) | 0.08 (9) | 0.408 (4) | 0.1406 (8) | 0.195 (6) | 0.1625 (7) | **0.815 (1)** |
| DAMI_Waveform | 0.7761 (3) | 0.2025 (6) | 0.3425 (5) | 0.175 (7) | 0.0338 (9) | 0.6528 (4) | 0.85 (2) | 0.0575 (8) | **0.975 (1)** |
| DAMI_WBC | 0.5522 (5) | 0.4175 (6) | **0.99 (1)** | 0.01 (9) | 0.9284 (3) | 0.9653 (2) | 0.195 (7) | 0.1775 (8) | 0.765 (4) |
| DAMI_WDBC | 0.1592 (8) | 0.6575 (3) | 0.755 (2) | 0.6575 (3) | 0.1095 (9) | **0.8368 (1)** | 0.235 (6) | 0.6575 (3) | 0.2 (7) |
| DAMI_Wilt | 0.4378 (4) | 0.645 (3) | 0.185 (7) | **1 (1)** | 0.005 (9) | 0.4045 (5) | 0.2 (6) | 0.995 (2) | 0.17 (8) |
| DAMI_WPBC | 0.2886 (8) | 0.695 (4) | 0.7575 (2) | 0.54 (6) | 0.2527 (9) | 0.3385 (7) | 0.7275 (3) | **0.95 (1)** | 0.615 (5) |
| ODDS_annthyroid | 0.4428 (8) | 0.2525 (9) | 0.58 (6) | **1 (1)** | 0.9343 (2) | 0.7743 (3) | 0.485 (7) | 0.61 (5) | 0.7475 (4) |
| ODDS_arrhythmia | 0.4328 (6) | 0.4425 (5) | 0.165 (9) | 0.94 (2) | 0.7055 (3) | 0.6927 (4) | 0.205 (7) | 0.19 (8) | **0.99 (1)** |
| ODDS_breastw | 0.6318 (6) | 0.0975 (8) | 0.8 (3) | 0.7025 (4) | **1 (1)** | 0.0556 (9) | 0.185 (7) | 0.8675 (2) | 0.7025 (4) |
| ODDS_glass | 0.5473 (3) | 0.5175 (4) | 0.105 (8) | 0.325 (7) | 0.0229 (9) | 0.3524 (6) | 0.91 (2) | **0.9825 (1)** | 0.4428 (5) |
| ODDS_ionosphere | 0.3781 (5) | 0.5125 (4) | **0.985 (1)** | 0.77 (3) | 0.205 (7) | 0.0694 (8) | 0.055 (9) | 0.2825 (6) | 0.9403 (2) |
| ODDS_letter | 0.5224 (5) | 0.7425 (3) | 0.055 (7) | 0.005 (8) | 0.005 (9) | **0.9722 (1)** | 0.95 (2) | 0.44 (6) | 0.54 (4) |
| ODDS_lympho | 0.4428 (6) | 0.8275 (2) | 0.03 (9) | 0.3 (7) | **0.9721 (1)** | 0.8108 (3) | 0.6375 (5) | 0.2425 (8) | 0.805 (4) |
| ODDS_mammography | 0.4129 (6) | 0.4175 (5) | 0.9075 (2) | 0.215 (7) | **1 (1)** | 0.6563 (4) | 0.045 (9) | 0.1825 (8) | 0.6775 (3) |
| ODDS_mnist | 0.5025 (5) | 0.4175 (6) | 0.24 (8) | 0.055 (9) | 0.7522 (2) | 0.6302 (3) | 0.505 (4) | 0.405 (7) | **0.79 (1)** |
| ODDS_musk | 0.7662 (2) | 0.6475 (4) | 0.59 (5) | 0.155 (8) | **1 (1)** | 0.066 (9) | 0.675 (3) | 0.3575 (7) | 0.3881 (6) |
| ODDS_optdigits | 0.5572 (7) | 0.8925 (3) | 0.3225 (9) | **1 (1)** | 0.6299 (6) | 0.7465 (5) | 0.765 (4) | 0.905 (2) | 0.45 (8) |
| ODDS_pendigits | 0.6219 (8) | 0.6625 (6) | 0.6275 (7) | 0.1725 (9) | **1 (1)** | **1 (1)** | 0.675 (5) | 0.7525 (4) | 0.77 (3) |
| ODDS_pima | 0.4776 (6) | 0.4225 (7) | **0.9525 (1)** | 0.92 (2) | 0.3775 (8) | 0.5729 (5) | 0.05 (9) | 0.8875 (3) | 0.86 (4) |
| ODDS_satellite | 0.408 (4) | 0.2725 (8) | 0.36 (7) | 0.905 (3) | **1 (1)** | 0.4063 (5) | 0.25 (9) | 0.365 (6) | 0.985 (2) |
| ODDS_satimage-2 | 0.5771 (6) | 0.3325 (8) | 0.7725 (5) | **0.98 (1)** | 0.855 (4) | 0.5226 (7) | 0.94 (2) | 0.915 (3) | 0.005 (9) |
| ODDS_speech | 0.8209 (3) | 0.8275 (2) | 0.57 (5) | 0.3275 (8) | **0.8945 (1)** | 0.349 (7) | 0.025 (9) | 0.42 (6) | 0.6825 (4) |
| ODDS_thyroid | 0.4925 (7) | 0.2925 (8) | 0.925 (3) | **1 (1)** | 0.999 (2) | 0.5556 (6) | 0.24 (9) | 0.58 (5) | 0.91 (4) |
| ODDS_vertebral | 0.3881 (6) | 0.6975 (4) | 0.045 (9) | 0.135 (8) | 0.2418 (7) | 0.8715 (2) | 0.7375 (3) | **0.885 (1)** | 0.49 (5) |
| ODDS_vowels | 0.403 (5) | 0.2875 (7) | 0.385 (6) | 0.63 (3) | 0.0259 (9) | 0.8403 (2) | 0.055 (8) | 0.515 (4) | **0.9652 (1)** |
| ODDS_wbc | 0.3731 (6) | 0.5575 (4) | 0.435 (5) | 0.19 (9) | **0.9771 (1)** | 0.7326 (3) | 0.2 (8) | 0.32 (7) | 0.9478 (2) |
| ODDS_wine | 0.3632 (5) | 0.5875 (2) | 0.44 (4) | 0.055 (8) | 0.1512 (7) | **0.6806 (1)** | 0.21 (6) | 0.0425 (9) | 0.5275 (3) |
| Average | 0.4811 (7) | 0.5001 (6) | 0.5658 (3) | 0.4565 (8) | 0.5829 (2) | 0.5615 (4) | 0.4379 (9) | 0.5034 (5) | **0.6945 (1)** |
| STD | 0.1354 | 0.1911 | 0.3005 | 0.3659 | 0.3996 | 0.2903 | 0.3397 | 0.3122 | 0.2457 |
| Avg. Rank | 5.5641 | 4.9744 | 4.7692 | 5.5897 | 4.5897 | 4.7179 | 5.6667 | 5.2564 | 3.6667 |

Table D5: Method evaluation on iForest (normalized AP rank). The best performing method is highlighted in bold. The algo. rank is provided in parenthesis (lower ranks denote better performance). HPOD achieves the best performance among all.

| datasets | Random | GB | ISAC | AS | HyperEns | MetaOD | Default | HPOD_0 | HPOD |
|---|---|---|---|---|---|---|---|---|---|
| DAMI_ALOI | 0.3979 (3) | 0.0191 (8) | 0.5208 (2) | 0.0868 (7) | 0.2789 (5) | 0.1319 (6) | 0.0035 (9) | 0.359 (4) | **0.7326 (1)** |
| DAMI_Annthyroid | 0.654 (6) | **0.9583 (1)** | 0.0764 (9) | 0.7535 (5) | 0.5405 (7) | 0.9236 (3) | 0.8789 (4) | 0.5167 (8) | 0.9358 (2) |
| DAMI_Arrhythmia | 0.4879 (6) | **0.8924 (1)** | 0.4427 (7) | 0.7517 (4) | 0.8166 (2) | 0.2188 (8) | 0.1107 (9) | 0.6146 (5) | 0.8125 (3) |
| DAMI_Cardiotocography | 0.481 (6) | 0.691 (2) | **0.9115 (1)** | 0.6319 (3) | 0.5765 (4) | 0.4583 (7) | 0.5433 (5) | 0.4115 (9) | 0.4444 (8) |
| DAMI_Glass | 0.6021 (3) | 0.2517 (8) | 0.4253 (6) | 0.3993 (7) | 0.5606 (4) | 0.1684 (9) | **0.9585 (1)** | 0.5396 (5) | 0.7795 (2) |
| DAMI_HeartDisease | 0.526 (8) | 0.8681 (3) | 0.9028 (2) | 0.6997 (5) | 0.6042 (7) | 0.6319 (6) | 0.09 (9) | 0.8326 (4) | **0.941 (1)** |
| DAMI_InternetAds | 0.4498 (7) | 0.8438 (3) | 0.2465 (8) | 0.0174 (9) | 0.872 (2) | **0.9132 (1)** | 0.4602 (6) | 0.6438 (5) | 0.7543 (4) |
| DAMI_PageBlocks | 0.3183 (7) | 0.0972 (8) | 0.6649 (2) | 0.3333 (6) | 0.5066 (5) | 0.5903 (4) | 0.0138 (9) | 0.6167 (3) | **0.9688 (1)** |
| DAMI_PenDigits | 0.481 (4) | 0.026 (9) | 0.2951 (6) | **0.9878 (1)** | 0.4976 (3) | 0.8924 (2) | 0.0623 (8) | 0.3108 (5) | 0.1806 (7) |
| DAMI_Pima | 0.3599 (7) | 0.0677 (9) | 0.9583 (2) | 0.5417 (6) | 0.6083 (5) | 0.783 (3) | 0.1211 (8) | 0.6215 (4) | **0.9861 (1)** |
| DAMI_Shuttle | 0.4844 (5) | 0.092 (9) | 0.5035 (4) | 0.3472 (6) | 0.5654 (3) | 0.309 (7) | 0.1886 (8) | **0.7326 (1)** | 0.6817 (2) |
| DAMI_SpamBase | 0.5433 (6) | 0.8958 (4) | 0.9028 (3) | 0.9479 (2) | 0.5772 (5) | 0.1823 (9) | 1 (1) | 0.2563 (7) | 0.218 (8) |
| DAMI_Stamps | 0.6298 (5) | 0.8941 (2) | 0.434 (8) | 0.7795 (4) | 0.6007 (6) | 0.1406 (9) | 0.564 (7) | 0.8868 (3) | **0.9913 (1)** |
| DAMI_Waveform | 0.4844 (6) | 0.0799 (8) | **0.9757 (1)** | 0.3212 (7) | 0.546 (5) | 0.6528 (4) | 0.0138 (9) | 0.8441 (3) | 0.872 (2) |
| DAMI_WBC | 0.3737 (7) | 0.1372 (9) | 0.7569 (4) | 0.7917 (2) | 0.5273 (6) | **0.9653 (1)** | 0.1592 (8) | 0.7563 (5) | 0.7917 (2) |
| DAMI_WDBC | 0.526 (8) | 0.7188 (4) | 0.6354 (7) | 0.6389 (6) | 0.5038 (9) | 0.8368 (2) | 0.6903 (5) | 0.7955 (3) | **0.901 (1)** |
| DAMI_Wilt | 0.5467 (4) | 0.2882 (8) | 0.3264 (6) | 0.3264 (6) | 0.5827 (3) | 0.4045 (5) | 0.1125 (9) | **0.8028 (1)** | 0.7889 (2) |
| DAMI_WPBC | 0.4983 (3) | 0.0799 (8) | 0.2656 (7) | 0.0313 (9) | 0.4997 (2) | 0.3385 (6) | 0.4273 (5) | 0.4399 (4) | **0.7014 (1)** |
| OODS_annthyroid | 0.5952 (8) | 0.6528 (7) | 0.7292 (3) | 0.6667 (5) | 0.5661 (9) | 0.7743 (2) | 0.6713 (4) | **0.8278 (1)** | 0.6574 (6) |
| OODS_arrhythmia | 0.4706 (6) | 0.3958 (7) | 0.5399 (5) | 0.0174 (9) | 0.6879 (3) | 0.6927 (2) | 0.2664 (8) | 0.5816 (4) | **0.8564 (1)** |
| OODS_breastw | 0.4152 (7) | 0.599 (5) | 0.7257 (2) | 0.5104 (6) | **0.8844 (1)** | 0.0556 (9) | 0.6609 (3) | 0.6003 (4) | 0.4115 (8) |
| OODS_glass | 0.564 (5) | 0.1684 (8) | 0.75 (4) | 0.0729 (9) | 0.517 (6) | 0.3524 (7) | **0.9308 (1)** | 0.7594 (3) | 0.9201 (2) |
| OODS_ionosphere | 0.3218 (5) | 0.0451 (8) | 0.1875 (6) | **0.941 (1)** | 0.4637 (4) | 0.0694 (7) | 0.0035 (9) | 0.85 (3) | **0.941 (1)** |
| OODS_letter | 0.4983 (6) | 0.224 (7) | 0.1354 (8) | 0.7778 (4) | 0.5384 (5) | **0.9722 (1)** | 0.0104 (9) | 0.7812 (3) | 0.8201 (2) |
| OODS_lympho | 0.4152 (5) | 0.0556 (9) | 0.2951 (6) | 0.6233 (4) | 0.7772 (2) | **0.8108 (1)** | 0.6522 (3) | 0.2455 (7) | 0.1597 (8) |
| OODS_mammography | 0.5502 (7) | 0.6458 (4) | **0.7361 (1)** | 0.5521 (6) | 0.5356 (8) | 0.6563 (3) | 0.7076 (2) | 0.5267 (9) | 0.5536 (5) |
| OODS_mnist | 0.4187 (6) | 0.1285 (8) | 0.8507 (2) | 0.2917 (7) | 0.618 (5) | 0.6302 (4) | 0.0035 (9) | 0.7674 (3) | **0.9152 (1)** |
| OODS_musk | 0.2595 (8) | 0.9253 (2) | **0.9253 (1)** | 0.3368 (7) | 0.7661 (5) | 0.066 (9) | 0.8443 (4) | 0.6722 (6) | 0.9239 (3) |
| OODS_optdigits | 0.4879 (7) | 0.6215 (4) | 0.6597 (3) | 0.4861 (8) | 0.6118 (5) | 0.7465 (2) | 0.6003 (6) | 0.4281 (9) | **0.9325 (1)** |
| OODS_pendigits | 0.526 (5) | 0.0764 (9) | 0.9826 (2) | 0.3611 (7) | 0.6893 (4) | 1 (1) | 0.8166 (3) | 0.2007 (8) | 0.3837 (6) |
| OODS_pima | 0.3737 (7) | 0.3802 (6) | 0.4514 (4) | **0.8628 (1)** | 0.609 (2) | 0.5729 (3) | 0.1246 (9) | 0.4038 (5) | 0.2535 (8) |
| OODS_satellite | 0.4533 (6) | 0.9861 (2) | 0.3247 (8) | 0.2535 (9) | 0.51 (5) | 0.4063 (7) | **0.9965 (1)** | 0.5608 (4) | 0.7274 (3) |
| OODS_satimage-2 | 0.436 (3) | 0.3854 (5) | 0.0069 (9) | 0.4132 (4) | **0.7848 (1)** | 0.5226 (2) | 0.0138 (8) | 0.266 (6) | 0.191 (7) |
| OODS_speech | 0.6817 (3) | 0.6458 (4) | 0.5573 (5) | 0.5573 (5) | **0.9052 (1)** | 0.349 (7) | 0.0069 (9) | 0.259 (8) | 0.7024 (2) |
| OODS_thyroid | 0.5294 (7) | **0.9549 (1)** | 0.2309 (9) | 0.8646 (3) | 0.4976 (8) | 0.5556 (6) | 0.7197 (5) | 0.7222 (4) | 0.8819 (2) |
| OODS_vertebral | 0.6367 (7) | 0.8333 (6) | 0.9549 (3) | 0.9757 (2) | 0.3239 (8) | 0.8715 (5) | 0.0208 (9) | 0.9125 (4) | **0.9861 (1)** |
| OODS_vowels | 0.5606 (4) | 0.2639 (8) | 0.3038 (7) | **0.8785 (1)** | 0.6692 (3) | 0.8403 (2) | 0.0208 (9) | 0.4854 (6) | 0.5382 (5) |
| OODS_wbc | 0.4844 (6) | 0.1389 (7) | 0.559 (5) | 0.1007 (8) | 0.5599 (4) | 0.7326 (3) | **0.9308 (1)** | 0.8899 (2) | 0.0313 (9) |
| OODS_wine | 0.5917 (4) | **0.9063 (1)** | 0.059 (8) | 0.0104 (9) | 0.501 (6) | 0.6806 (2) | 0.5709 (5) | 0.6031 (3) | 0.3875 (7) |
| Average | 0.4901 (7) | 0.4598 (8) | 0.5438 (5) | 0.5113 (6) | 0.5969 (3) | 0.5615 (4) | 0.4095 (9) | 0.5981 (2) | **0.6835 (1)** |
| STD | 0.0960 | 0.3493 | 0.2904 | 0.3025 | 0.1364 | 0.2903 | 0.3613 | 0.2087 | 0.2790 |
| Avg. Rank | 5.7179 | 5.6923 | 4.7692 | 5.3846 | 4.5641 | 4.5385 | 6.0769 | 4.6410 | 3.5128 |

### D.6 Case Study on How HPOD Adaptively Finds Better HPs

We trace how HPOD identifies better HPs over iterations. Example of tuning LOF on `Cardiotocography` dataset is provided in Table D6. Among 200 candidate HP settings (see Appx. Table D2b), the optimal HP setting is {'Chebyshev', 79} with AP=0.3609. In 30 iterations, HPOD gradually identifies better HPs (closer to optimal), i.e., {'Chebyshev', 73}. Its AP improves from 0.2866 (1-$st$ iteration) to 0.357 (30-$th$ iteration).

### D.7 Ablation Studies and Other Analysis

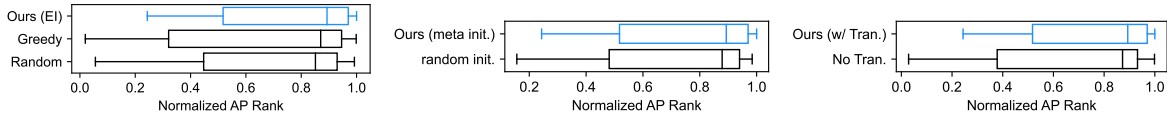

(a) Ablation of acquisition  (b) Ablation of initialization  (c) Ablation of surrogate transfer

Figure D1: (*a*) Ablation of EI (med.=0.893) vs. the greedy (med.=0.870) and random acquisition (med.=0.851) (*b*) Ablation of meta- (med.=0.893) vs. random-initialization (med. =0.874) of the surrogate function (*c*) ours w/ surrogate transfer (med.=0.893) vs. without transfer (med. =0.872).

**The Choices of Acquisition Function**. HPOD uses the EI acquisition to select an HP based on the surrogate function's prediction (see §3.4.2). We compare it with the random and greedy acquisition (latter picks the HP with the highest predicted performance, ignoring uncertainty) in Fig. D1a, where EI-based acquisition performs best.

**Surrogate Initialization**. HPOD uses meta-learning to initialize the surrogate (see §3.5.1). Fig. D1b shows its advantage over random initialization with higher performance.

**The Effect of Surrogate Transfer**. To improve the prediction performance of the surrogate function, HPOD transfers meta-surrogate functions from similar meta-train tasks (§3.5.2). Fig. D1c shows the transfer helps find better HPs, demonstrating the added value of meta-learning besides PPE training of $f(\cdot)$ and surrogate initialization.

Table D6: Full trace of HPOD on `Cardiotocography` dataset. Over iterations (col. 1), HPOD gradually identifies better HPs (col. 2 &3), with higher AP (col. 4). The optimal HP on from the meta-HP set is {'Chebyshev', 79}, which HPOD gets closer to the optimal HP during its adaptive search (i.e., finding {'Chebyshev', 73} in 30 iterations).

| # Iter | Metrics | # Neighbors | Norm. AP Rank |
|---|---|---|---|
| 1 | Manhattan | 23 | 0.2866 |
| 2 | Manhattan | 23 | 0.2866 |
| 3 | Manhattan | 23 | 0.2866 |
| 4 | Manhattan | 23 | 0.2866 |
| 5 | Cosine | 41 | 0.327 |
| 6 | Cosine | 42 | 0.327 |
| 7 | Cosine | 55 | 0.3438 |
| 8 | Cosine | 55 | 0.3438 |
| 9 | Cosine | 55 | 0.3438 |
| 10 | Cosine | 55 | 0.3438 |
| 11 | Cosine | 55 | 0.3438 |
| 12 | Cosine | 55 | 0.3438 |
| 13 | Cosine | 55 | 0.3438 |
| 14 | Chebyshev | 72 | 0.3569 |
| 15 | Chebyshev | 72 | 0.3569 |
| 16 | Chebyshev | 72 | 0.3569 |
| 17 | Chebyshev | 72 | 0.3569 |
| 18 | Chebyshev | 72 | 0.3569 |
| 19 | Chebyshev | 72 | 0.3569 |
| 20 | Chebyshev | 72 | 0.3569 |
| 21 | Chebyshev | 72 | 0.3569 |
| 22 | Chebyshev | 73 | 0.357 |
| 23 | Chebyshev | 73 | 0.357 |
| 24 | Chebyshev | 73 | 0.357 |
| 25 | Chebyshev | 73 | 0.357 |
| 26 | Chebyshev | 73 | 0.357 |
| 27 | Chebyshev | 73 | 0.357 |
| 28 | Chebyshev | 73 | 0.357 |
| 29 | Chebyshev | 73 | 0.357 |
| 30 | Chebyshev | 73 | 0.357 |
| 31 | Chebyshev | 73 | 0.357 |
| 32 | Chebyshev | 73 | 0.357 |
| 33 | Chebyshev | 73 | 0.357 |
| 34 | Chebyshev | 73 | 0.357 |
| 35 | Chebyshev | 73 | 0.357 |
| 36 | Chebyshev | 73 | 0.357 |
| 37 | Chebyshev | 73 | 0.357 |
| 38 | Chebyshev | 73 | 0.357 |
| 39 | Chebyshev | 73 | 0.357 |
| 40 | Chebyshev | 73 | 0.357 |
| 41 | Chebyshev | 73 | 0.357 |
| 42 | Chebyshev | 73 | 0.357 |
| 43 | Chebyshev | 73 | 0.357 |
| 44 | Chebyshev | 73 | 0.357 |
| 45 | Chebyshev | 73 | 0.357 |
| 46 | Chebyshev | 73 | 0.357 |
| 47 | Chebyshev | 73 | 0.357 |
| 48 | Chebyshev | 73 | 0.357 |
| 49 | Chebyshev | 73 | 0.357 |
| 50 | Chebyshev | 73 | 0.357 |
| **Optimal** | Chebyshev | 79 | **0.3609** |

