# OpenReview forum: "HPOD: Hyperparameter Optimization for Unsupervised Outlier Detection"
_automl.cc/AutoML/2024/Conference — AutoML 2024_

### Official Review · Reviewer_MDdp · 2024-03-25

**Potential Impact On The Field Of Automl:** Not useful due to generalizability co…
**Potential Impact On The Field Of Automl Rating:** 2
**Technical Quality And Correctness Rating:** 2
**Clarity:** It is not a clearly written paper.
**Clarity Rating:** 1

**Summary Of Contributions:**

The paper proposes a novel approach towards AutoML forunsupervised Outlier Detection. The paper proposes an approach to perform  HPO in OD. It's a novel approach fora non trivial problem. SMBO is a well studies approach for HPO, and the authors adopt this.
It is of note that unlike other meta-learning based methods, HPOD works with both discrete and continuous hyperparameters. (although their approach to discretization  of continuous spaces makes this a vacuous claim)

**Actions Required To Increase Overall Recommendation:**

Completely reworking the intuition and the meta-learning approach. Also the paper should be made less of a chore to read, and organization be better.

**Overall Review:**

Positives:

1. This is a good paper, in terms of how it adresses a practical and important problem. The formulation is mostly robust and well thought out. It is fairly well presented.
2. The meta learning algorithm for PPE is interestingh in th eoffline phase, and the approach to use SMBO makes sense in the online phase.
3. Fairly comprehensive experimental section.


Negatives:
1. In OD is that the true labels are difficult to determine, and they are best described as pseudo labels. The assumption that these anomalies cover all possible out of distribution behavior is less than well supported.
2. The paper makes some strong claims. While the performance gains shown are somewhat convincing --- the generalizability is strongly dependent on data. Anomaly patters, are subjective and application dependent.
3. The overall approach is very complex, and number of moving parts is rather large.
4. If the model is not selected, but only the hyperparameters, that may be detrimental to OD.
5. Meta learning based approaches suffer from the limitation of metadata. So PPE is limited by the datasets used. How is similarity of datasets determined ? There is sufficient prior work where meta-features do not work as expected. So this approach seems not very principled.
6. Paper is rather poorly organized

**Review Confidence:**

4

**Review Rating:**

4

**Review Summary:**

The authors present something novel. But it is poorly organized, and hides behind a wall of complexity. It is a chore to read the paper, and there are  many gaps in the very optimistic claims made by the paper.

**Technical Quality And Correctness:**

Fairly detailed presentation in terms of the technical details. Underlying idea is however less than principled or intuitive.

---

### Official Review · Reviewer_naqP · 2024-03-27

**Potential Impact On The Field Of Automl Rating:** 2
**Technical Quality And Correctness Rating:** 4
**Clarity Rating:** 3

**Summary Of Contributions:**

This work proposes a method for hyperparameter optimization for unsupervised outlier detection algorithms. Their method assumes access to labeled data that, in conjunction with a set of pre-selected hyperparameters, is used to create a dataset of outlier detection performances per hyperparameter setting. This dataset is used to train a performance evaluator to predict the performance of unlabeled data. Furthermore, it is used to initialize a surrogate model. Bayesian optimization is then used to find a new hyperparameter setting that the performance evaluator evaluates.

**Actions Required To Increase Overall Recommendation:**

* The clarity of the paper could be improved by a schematic diagram that shows the interplay of the different components of the algorithm.
* I'm lacking a strong motivation for the necessity of the discretization of  $\mathbf{\lambda}_{\textrm{meta}}$

**Clarity:**

The overall description is clear. However, the paper sometimes reads as if the authors were struggling with the page limit. For example, words in the running text are shortened (e.g., "perf." instead of "performance").

The paper would also benefit from a schematic diagram that sketches the approach. The overall idea is straightforward but it's not always easy to parse the individual components from the text.

**Overall Review:**

**Strengths**

The paper presents a novel approach to hyperparameter optimization for unsupervised outlier detection. Overall, the algorithmic design is sound and well-motivated. The paper is generally clear, including the easy-to-parse figures.

**Questions and weaknesses**

I do not understand the motivation for discretizing $\mathbf{\lambda}_{\textrm{meta}}$. It looks like this is done to make the optimization of EI feasible. However, there are gradient-based approaches that don't require a finite set of candidates.  Furthermore, the paper lacks a section on limitations.

**Potential Impact On The Field Of Automl:**

The method itself is an AutoML method but is tailored toward unsupervised outlier detection tasks. Thus, it is relevant but its generalizability is limited.

**Review Confidence:**

2

**Review Rating:**

7

**Review Summary:**

Overall, the paper proposes a novel algorithm for hyperparameter optimization for unsupervised outlier detection. While the impact of the work is limited to this setting, the algorithmic design is reasonable and gives good empirical performance.

**Technical Quality And Correctness:**

The technical descriptions are detailed and the paper features a comprehensive empirical evaluation.

---

### Official Review · Reviewer_seJT · 2024-03-27

**Potential Impact On The Field Of Automl Rating:** 3
**Technical Quality And Correctness Rating:** 4
**Clarity Rating:** 3

**Summary Of Contributions:**

The paper provides a version of sequential model-based optimisation that can be used for hyperparameter optimisation for unsupervised outlier detection methods. They do this by combining results from many supervised outlier detection data sets to learn a surrogate model of performance, which they can use as a proxy for evaluating the models on the unsupervised outlier detection task.

**Actions Required To Increase Overall Recommendation:**

See above responses. Especially, I would like a clarification of the method's own hyperparameters.

**Clarity:**

The paper is generally clearly written. The followinf are smaller clarifications, and questions.

In the introduction, the argument is made that HPO is needed because there is great variation in performance between HPs for a data set, but isn't the interesting question whether there is variation in what HP is best between data sets?

Lines 59 and 66: is HPOD better than all or most of the baselines?
You should add a citation to your description of initialising BO with good HPs from other tasks, as this is not a new contribution.

In related work, I could not follow the description of the second group (lines 77-79).

Small:
- HyperParameter Optimisation for unsupervised OD problem adds up to (HPOOD), not HPOD. Consider changing the underlining in line 42.
- In Fig 1, b) uses different symbols to a) and c) (see GB).
- Ideally, add citations to table 1.
- Line 325: two what?
- The caption for 4.2.2 bleeds into the margin.

**Overall Review:**

This is a well-written paper that puts a lot of different building blocks together to compose a BO-based approach for HPO for unsupervised outlier detection. There are some smaller improvements that could be made to the clarity.

The paper would be improved by a reflection of potential harms and benefits in the broader impact statement. There are many potential uses of unsupervised outlier detection.

**Potential Impact On The Field Of Automl:**

This paper seems to have the potential for medium impact, as a step towards better unsupervised outlier detection algorithms.

**Reproducibility:**

Where is the amount of compute used listed? How long does the preprocessing take?

**Review Confidence:**

3

**Review Rating:**

8

**Review Summary:**

A well-written paper with a clear application which could be adopted through AutoML for a variety of problems.

**Technical Quality And Correctness:**

If you compare expected improvement to other selection criteria, it would be more informative to compare it to other established acquisition functions, e.g. probability of improvement and upper confidence bound.

Lines 318-320: What are HPOD's hyperparameters?

---

### Meta-Review · Area_Chair_n2pk · 2024-04-22

**Paper Recommendation:** Accept
**Confidence:** 4

**Metareview:**

The authors introduce a novel HPO method in unsupervised outlier detection algorithms. The approach leverages a combination of labeled data and pre-selected hyperparameters to train a performance evaluator and initialize a surrogate model.

Most reviewers lean towards (weak) acceptance, with the exception of one reviewer who expressed concerns about generalizability. As these concerns were addressed by the authors in the rebuttal, who have updated the paper accordingly, I recommend acceptance.

---

### Decision · Program_Chairs · 2024-04-29

**Decision:**

Accept

**Comment:**

Thank you for submitting your paper. We are happy to tell you that we accept your paper to the main track. See you in Paris.